# Asian monsoon rainfall variation during the Pliocene forced by global temperature change

Hanlin Wang [1], Huayu Lu [1,2]*, Lin Zhao [1], Hongyan Zhang [1], Fang Lei[1] & Yichao Wang[1]

The Asian monsoon variations under global temperature changes during the Pliocene are still debated. Here we use a sedimentary record of phytoliths (plant silica) from the Weihe Basin, central China, to explore the history of $C_4$ grasses and quantitatively reconstruct the Asian monsoon climate since the late Miocene. Our results show that $C_4$ grasses have been a dominant grassland component since ~11.0 Ma. A subsequent marked decrease in warm- and humid-adapted $C_4$ grasses and an increase in cool- and dry-adapted $C_3$ grasses occurred in the Pliocene, ~4.0 Ma; the phytolith-based quantitative reconstruction of mean annual pre-cipitation marked a decrease from 800~1673 mm to 443~900 mm, indicating a reduction in Asian monsoon rainfall in the Pliocene. Our newly obtained records conflict with the hypothesis that the growth of the Tibetan Plateau strengthened the Asian monsoon rainfall. Nevertheless, they emphasize the importance of global temperature as a determinant of Pliocene Asian monsoon variations.

[1] School of Geography and Ocean Science, Nanjing University, Nanjing 210023, China. [2] Jiangsu Provincial Collaborative Innovation Centre of Climate Change, Nanjing 210023, China. *email: huayulu@nju.edu.cn

The Asian monsoon system, which controls precipitation over mainland Asia, is characterized by abundant summer precipitation brought by warm, moisture-bearing winds of maritime origin. The variability in Asian monsoon precipitation since the late Miocene has been widely investigated, but the findings remain controversial. A marked shift in the stable carbon isotopic composition ($\delta^{13}C$) of pedogenic carbonate from northern Pakistan suggests the expansion of $C_4$ grasses (i.e., plants that use the $C_4$ photosynthetic pathway) during the late Miocene, indicating the development of seasonal climate[1] and increased warm-season precipitation (WSP)[2]. However, one of the objections to increased WSP as an explanation, comes from evidence in the western Arabian Sea, in the form of reduced summer precipitation and/or increased winter precipitation, which are required to explain the increased $\delta D$ values of leaf wax[3]. A similar debate exists for the Chinese Loess Plateau (CLP), where $\delta^{13}C$ profiles of pedogenic carbonate suggest that an expansion of $C_4$ grasses occurred at ~4.0 Ma[4,5], which has been interpreted as increased WSP in East Asia[5]. However, an integrated study of the $\delta^{13}C$ records of pedogenic carbonate and tooth enamel suggested a reduction in WSP in East Asia at ~4.0 Ma[6].

A major reason for the discrepancy above is the use of stable carbon isotopic compositions as a proxy for the expansion of $C_4$ grasses and therefore monsoon precipitation, whereas the drivers of $C_4$ expansion and the environmental niches of $C_4$ grasses are still unclear. Thus, clarifying the environmental implications of $C_4$ grasses is essential to resolving the discrepancies regarding the changes in Asian monsoon precipitation since the late Miocene.

When comparing the environmental niches between grasses and trees, onset of a dry season and/or an increase in aridity, combined with increased fire frequency, could favour the growth of $C_3$ and $C_4$ grasses, and limit the growth of woody species[7,8]. Within grasses, a detailed comparison of the environmental niches of $C_3$ and $C_4$ grasses is needed. Recent work indicates that PACMAD species (PACMAD is the clade of grasses that includes the grass subfamilies Panicoideae, Aristidoideae, Chloridoideae, Centothecoideae, Micrairoideae, Arundinoideae and Danthonioideae) tend to be warm-adapted, irrespective of whether they are $C_4$ grasses, whereas the $C_3$ Pooideae tend to be cold-adapted[9]. Thus, cold tolerance may be as important as $C_3$ and $C_4$ differences in determining the distribution of grasses along temperature gradients[9]. On the other hand, environmental niches tend to vary within the different $C_4$ lineages, reflecting responses to precipitation[9] and elevated $CO_2$ levels[10]. For a $C_4$-dominated grass community, the environmental niches of specific clades of grasses that dominate grasslands are the key to determining the environmental niches of the entire grass community[8]. If it was possible to evaluate the ecological role (from non-dominant to dominant) played by each clade of grasses during the expansion of $C_4$ grasses, the corresponding environmental controls and driving mechanisms could be identified, which would eventually lead to a better understanding of the linkages between $C_4$ grasses and late Neogene climate change. For this reason, the study of phytolith assemblages as a direct record of late Neogene plant communities has attracted increasing interest for understanding the origin of $C_4$ expansion[11]. As an alternative to stable carbon isotope ($\delta^{13}C$) analysis, fossil phytoliths can provide detailed information on $C_3$ (forests/shrubs/$C_3$ grasslands) and $C_4$ (Chloridoideae-dominated/Panicoideae-dominated grasslands) vegetation[12], thus providing detailed information on climate change.

Here we explore the history of vegetation and the monsoon climate in East Asia using a continuous record of phytolith assemblages since the late Miocene from Lantian (34.20 °N, 109.24 °E) in Weihe Basin, central China (Fig. 1). A total of

133 samples were analysed for phytoliths, of which 87 contained phytoliths and 38 contained sufficiently well-preserved phytoliths (Supplementary Fig. 1) for our quantitative palaeoclimatic analysis (Supplementary Tables 1, 2). The ages of the samples span the past ~11.0 Ma with an uneven distribution in time along the profile (Supplementary Fig. 2). First, we build a species-climate database to explore the quantitative relationship between modern grasses and climate (see Methods). Then, we use the phytolith assemblages to analyse the vegetation and habitat type, including the tree cover, grass community composition and percentage of $C_4$ grasses. Finally, we integrate these two datasets to quantitatively reconstruct the Asian monsoon climate since the late Miocene. Our results show that the percentage of $C_4$ grasses markedly decreases by ~4.0 Myr ago, which reveals a distinct decrease in monsoon precipitation during the Pliocene in East Asia.

## Results

**Environmental niches of modern grasses**. The climate data extracted from all available geo-referenced grasses from China provided clear evidence that certain grass lineages have specialized in certain habitats (Fig. 2). In terms of temperature, Pooideae stand out and occupy the cool end of the spectrum, whereas other subfamilies are indistinguishable (Fig. 2), as suggested by either the mean annual temperature (MAT) or the temperature of the warmest or coldest months (Fig. 2 and Supplementary Fig. 3). In terms of precipitation, Pooideae also stands apart as inhabiting the driest environments of all subfamilies, as suggested by the mean annual precipitation (MAP) and the warmest month mean precipitation (WMMP, the monsoon season in Asia). Chloridoideae adapted to relatively more humid environments than Pooideae, but these environments were still drier than those associated with Panicoideae, Oryzoideae and Bambusoideae (Fig. 2). With respect to the seasonality of precipitation, all subfamilies are similar (Supplementary Fig. 3). Upon further investigation of the climate data and the distribution of species that dominated the modern grassland in China, we found that grassland had more specialized habitats (Supplementary Fig. 4). For example, the grasslands dominated by Pooideae species were mainly distributed in northwestern China with a MAT < 11 °C, except for one (dominated by *Deyeuxia arundinacea* with a MAT = 12.3 °C), whereas grasslands dominated by Panicoideae grasses were distributed in southeastern China with a MAT > 11 °C (*Bothriochloa ischaemum* with a minimum MAT = 11.2 °C). These data provide a solid foundation for the quantitative reconstruction of temperature and precipitation based on phytolith assemblages (Supplementary Figs. 3 and 4). The environmental niches of grasses in China are partly different from those of grasses globally[9], suggesting a more specialized grass habitat on a regional scale that favours the quantitative reconstruction of regional climate. However, to use this regional-scale grass habitat for quantitative reconstructing, we need to make the assumption that the same species have thrived since the late Miocene, and that the environmental niches of these grass subfamilies have not evolved.

**Vegetation and grass community**. Most of the 38 samples used for quantitative analysis did not contain palm or other forest indicator morphotypes (Supplementary Tables 1 and 2). The relative abundance of non-grass plants as a proxy for tree cover (FI-t) ranged from 0 to ~12% (Fig. 3a). All 38 samples were dominated by grass silica short cells (GSSCs) from many types of grasses; however, they were dominated by morphotypes typical of open habitats, which comprised 99~100% of the GSSCs. The GSSCs from open-habitat grasses consisted of a mixture of morphotypes typical of Pooideae ($C_3$) and PACMAD ($C_3$ and $C_4$)

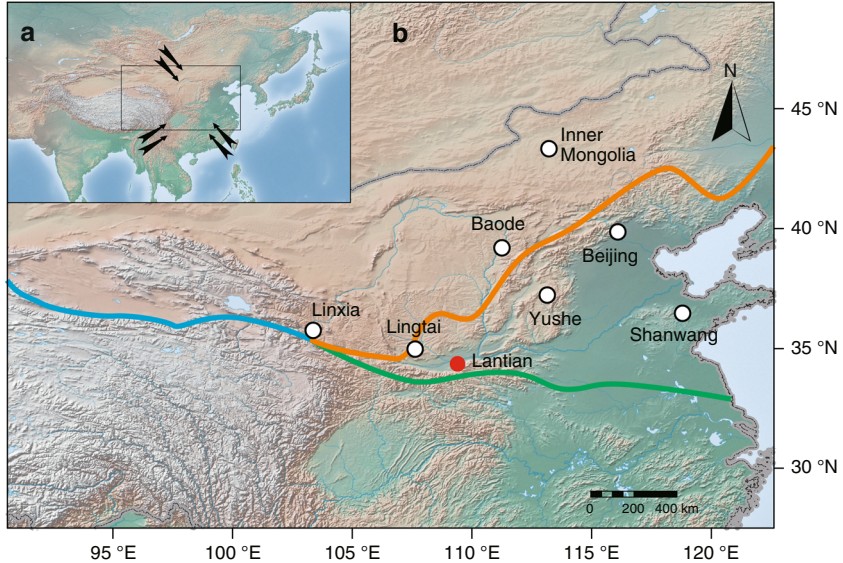

**Fig. 1** Climate of the study area and the vegetation distribution in East Asia. **a** Landform of Eurasia and the climate of East Asia; the prevailing modern atmospheric circulation patterns are indicated by arrows. **b** Localities referenced in the text and figures. Red dot: location of the study site; open black dots: locations mentioned in the text. Solid lines: boundaries between modern vegetation. Green line: the boundary between forest and grasses. Orange line: the boundary between $C_3$ grasses and $C_4$ grasses. Blue line: the boundary between modern desert steppe and alpine meadow. The distribution of modern vegetation is from the Chinese terrestrial ecosystem database (http://www.ecosystem.csdb.cn). The images of landform are from (https://maps.ngdc.noaa.gov/arcgis/rest/services).

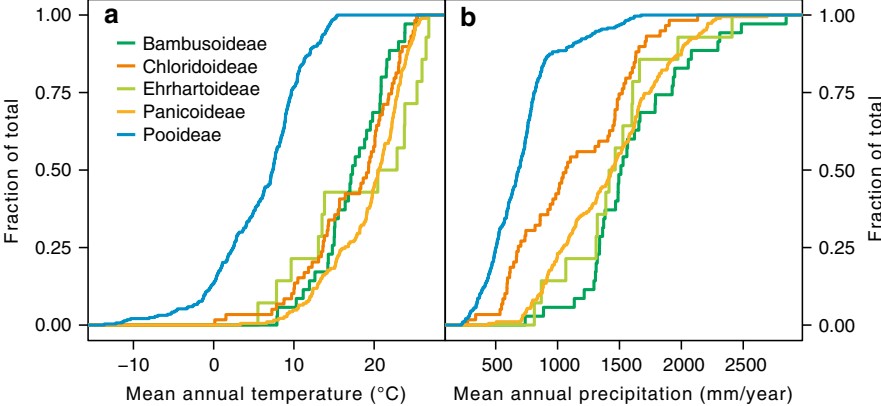

**Fig. 2** Species accumulation curves for the climate parameters in China, sorted by the major grass subfamilies. **a** Species accumulation curves for mean annual temperature. **b** Species accumulation curves for mean annual precipitation. These data represent 598 species records in the Flora of China (http://frps.iplant.cn/), while the collection localities are not necessarily in China. For data sources, see the Methods and Supplementary Note 2

grasses. Rather than an unchanging vegetation structure, the grass community exhibited significant variations over time (Fig. 3b, c): PACMAD grasses dominated the grass community (range 41~85%, average 71%) during 11.0~4.2 Ma. The minimum estimate for $C_4$ grasses was 25% (range 14~44%) and the maximum estimate was 75% (range 48~89%). Pooideae grasses began to dominate the grass community (range 50~91%, average 67%) during 4.2~2.6 Ma and the percentage of $C_4$ grasses decreased to a minimum of ~7% (range 1~11%) and a maximum of ~26% (range 8~42%). In the interval from 2.6 to ~0 Ma, our samples span only the last glacial–interglacial cycle and the Holocene (130~0 ka), when Pooideae grasses continued to dominate the grassland community (range 76~100%, average 93%) and the percentage of $C_4$ grasses reached to a minimum (range 0~15%, average 3%).

**Precipitation variations.** Phytoliths can be used to directly distinguish among many subfamilies of Poaceae[12]. Pooideae grasses

occupy the coldest and driest environment of all subfamilies in China (Fig. 2). Thus, the relative percentage of Pooideae grasses (in GSSCs) could be used as a qualitative proxy for temperature and precipitation (Fig. 4a). Based on the climate range of each subfamily, using the coexistence approach and ecosystem matching methods (see Methods), the reconstructed climate could be divided into three stages (Fig. 4e): first, a low percentage of Pooideae grasses, together with the presence of Bambusoideae and Oryzoideae during 11.0~4.2 Ma, indicated a warm and humid climate during the late Miocene. The reconstructed MAT and MAP were 11~15.3 °C and 800~1673 mm, respectively. The WMMP and the WSP (April to September) (which has fallen in the monsoon season and thus are indicators of monsoon precipitation) were 76~199 mm and 427~1197 mm, respectively. Second, Pooideae grasses started to increase and eventually dominated the grass community during 4.2~2.6 Ma, indicating that the climate became cooler and drier in the late Pliocene.

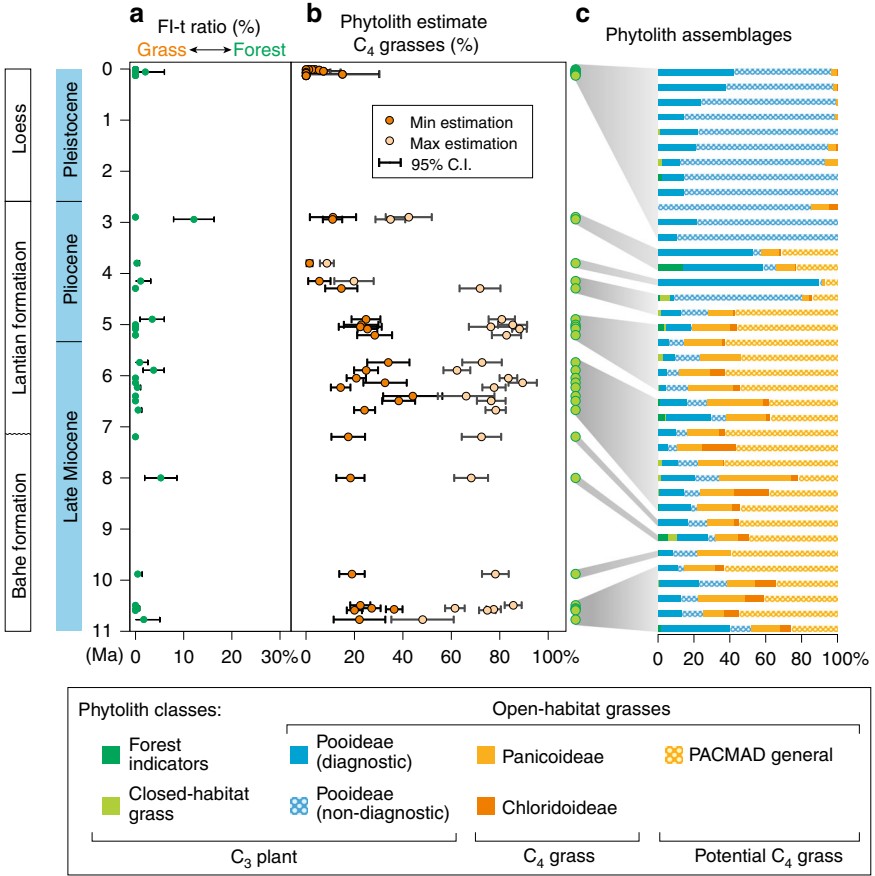

**Fig. 3** Phytolith records since ~11.0 Ma from Lantian, central China. **a** Tree cover estimated by phytolith assemblages by comparing forest indicator phytoliths (FI TOT) and diagnostic grass phytoliths (GSSC), FI-t ratio = FI TOT/(FI TOT + GSSC)%. **b** Potential $C_4$ grasses. The minimum estimation is calculated from PAN + CHLOR (Panicoideae + Chloridoideae) (Eq. 1) and the maximum estimation from PACMAD (a clade of grasses that includes the grass subfamilies Panicoideae–Aristidoideae–Chloridoideae–Centothecoideae–Micrairoideae–Arundinoideae–Danthoniodeae) (Eq. 2). The error bars are calculated from the confidence intervals (95% unconditional case) of each sample using the total count as the sample size. **c** Phytolith assemblages. For detailed information about these data and equations, see the Methods and Supplementary Table 2

However, the presence of Oryzoideae during this interval indicated that the climate was warmer and more humid than that during the Pleistocene. The reconstructed MAT and MAP decreased markedly to 9.7~11 °C and 812~900 mm, respectively. The WMMP and WSP were 76~130 mm and 417~540 mm, respectively. Third, the continued increase in Pooideae grasses, together with the absence of Bambusoideae and Oryzoideae during 2.6~0 Ma, indicated an even colder and drier climate during this interval. The reconstructed MAT and MAP decreased to 3.3~11 °C and 441~900 mm, respectively. The WMMP and WSP were 43~130 mm and 263~540 mm, respectively.

## Discussion
Our results reveal that $C_4$ grasses were moderately abundant (minimum estimate) to very abundant (maximum estimate) in the study area at ~11.0 Ma, which is most likely the earliest fossil record of $C_4$ grassland in East Asia. The record of $C_4$ grasses prior to this time is very sparse. Phytolith assemblages from Lower Miocene sediments in Shanwang in northern China indicate that potential $C_4$ grasses were present at very low frequencies, growing beneath the tree canopy or in a forest mosaic[13]. Stable carbon isotopic composition records from the South China Sea indicate that $C_4$ plants gradually appeared as a component of land vegetation beginning in the early Miocene[14]. $C_4$ plants were also a vegetation component in the arid Asian interior in the middle

Miocene, as evidenced by n-alkane carbon isotopic data from the North Pacific[15].

Our direct fossil phytolith evidence indicates that grasses dominated the vegetation, and that there was a mixture of $C_3$ and $C_4$ grasses during 11.0~4.2 Ma. A grass-dominated ecosystem was distributed across a vast area of northern China since the late Miocene (~11.0 Ma), as evidenced by the synthesis of Neogene pollen records[16]. Fossil mammals and their distribution in the late Miocene also document this vegetation pattern[17]. There is evidence that $C_4$ grasses subsequently became abundant. For example, the stable carbon isotopic composition of tooth enamel reveals that herbivores fed largely on $C_4$ grasses in Chinese Loess Plateau at ~7.0 Ma[6] and in Inner Mongolia at ~7.5 Ma[18]. Although there are large discrepancies in the timing of the $C_4$ expansion, all the data suggest that, prior to the expansion of $C_4$ grasses in the late Miocene and Pliocene, the vegetation consisted of open grassland rather than closed forest, and a similar vegetation pattern was discovered in North America using a phytolith-based vegetation reconstruction[8,19].

Our data suggest that a marked decrease in $C_4$ grasses (both maximum and minimum estimates) occurred at ~4.2 Ma, which is consistent with the general decreasing trend in $C_4$ grasses suggested by the $\delta^{13}C$ values of tooth enamel (Supplementary Fig. 5). The percentage of $C_4$ grasses in Inner Mongolia largely decreased at ~4.0 Ma, resembling the modern vegetation that is dominated by cold-adapted $C_3$ grasses[18]. The vegetation of the

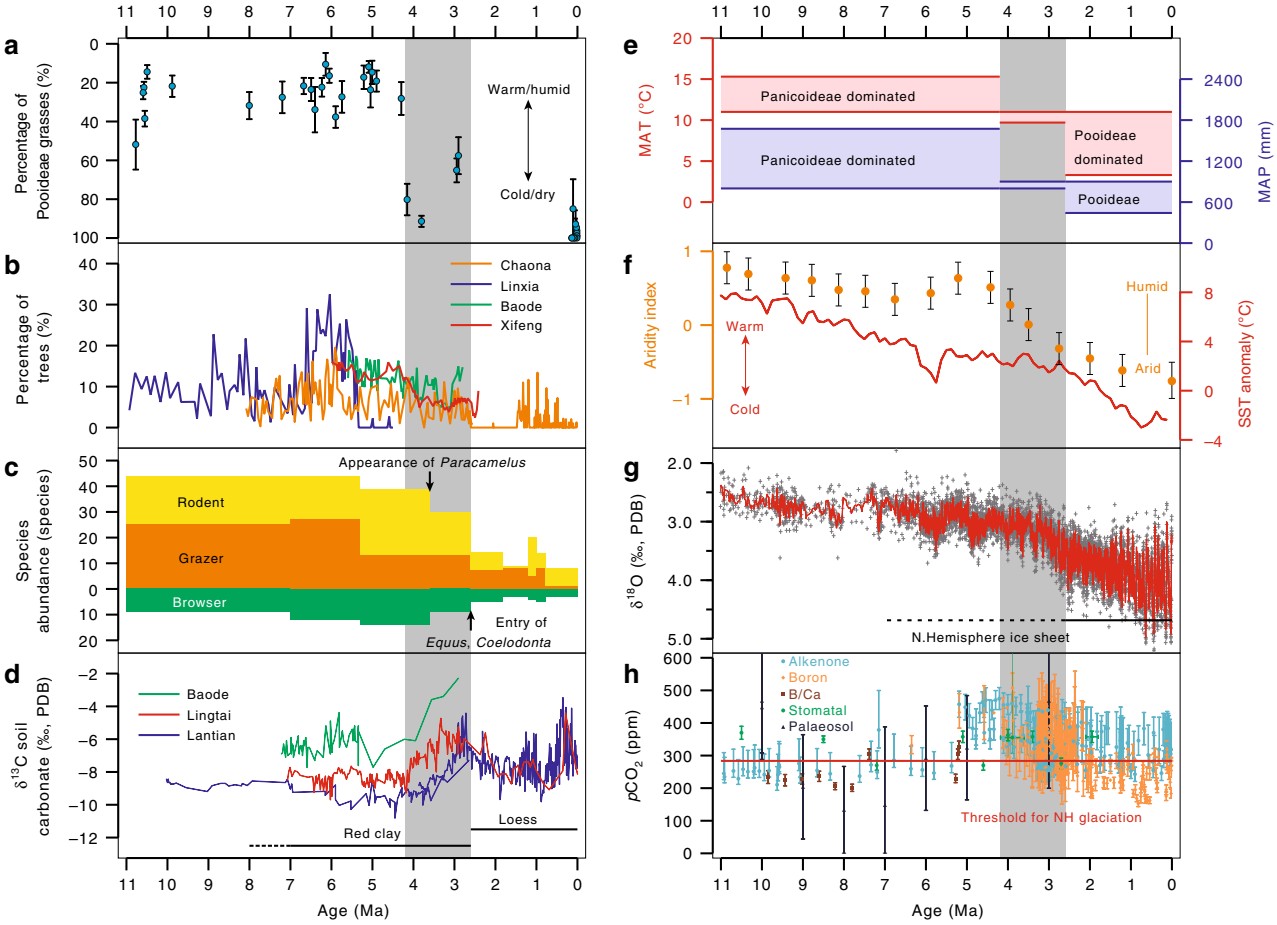

**Fig. 4** Reconstructed climate change in East Asia and the comparison with sea surface temperature, benthic oxygen isotope and atmospheric $CO_2$ level since ~11.0 Ma. **a** The percentage of Pooideae grasses estimated by phytolith assemblages, which were used as a proxy for temperature and/or precipitation. The error bars are calculated from the confidence intervals (95% unconditional case) of each sample using the total count as the sample size. **b** The percentage of trees derived from palynoflora in Xifeng[30], Baode[31], Linxia[32] and Chaona[33,34] as a percentage of broadleaf trees of palynoflora. **c** Species abundance of different types since the late Miocene in northern China. Browsers are large mammals with low-crowned teeth (brachydont) that lived in more humid environments and grazers are large mammals with high-crowned teeth (hypsodont) that lived in more arid conditions. Fossil mammal data of the late Miocene and Pliocene are from Wang et al.[17] and those of the Quaternary are from Xue et al.[37]. **d** Soil carbonate $\delta^{13}C$ records from Baode[6,57], Lingtai[4] and Lantian[5,22]. PDB, PeeDee Belemnite. **e** Phytolith-based quantitative reconstruction of mean annual temperature (MAT) and mean annual precipitation (MAP). **f** Synthesis of pollen and sedimentary records from northwest China to illustrate the evolution of aridity and the monsoon climate[46]; synthesis of sea surface temperature (SST) anomalies for the Northern Hemisphere (30–50°) based on $U^{k'}_{37}$ records from four locations[52]. **g** Benthic $^{18}O$ composite. **h** Synthesis of published $pCO_2$ proxy data obtained through a variety of different methodologies. For data sources, see Supplementary Note 3. The grey highlighted intervals indicate the Pliocene cooling (4.2–2.6 Ma) suggested by this study

Lingtai area also changed from $C_3/C_4$ mixed grassland in the late Miocene–Pliocene to pure $C_3$ grassland in the Quaternary, similar to modern vegetation[20]. In the Linxia Basin, $C_4$ grasses largely increased in the Quaternary, but they remained dominated by $C_3$ grasses, similar to modern vegetation[21].

A controversial issue is that our phytolith data provide a different estimate of $C_4$ grasses (both maximum and minimum estimates) from the estimate derived from $\delta^{13}C$ data of soil carbonates[5,22]. To test the validity of our estimate, we used modern soil phytolith assemblages from China[23] to calculate the percentage of $C_4$ grasses (maximum estimate); the result shows a pattern similar to the distribution of modern $C_4$ grasses (Supplementary Fig. 6). We also compared our phytolith data with the results of an integrated study of the $\delta^{13}C$ of bulk organic matter since the last glacial maximum[24]. It appears that our phytolith estimate (0~15%) underestimates the biomass of $C_4$ grasses (10~40%). Several researchers compared phytolith data and $\delta^{13}C$ values of soil organic matter in the Miocene and found a general consistency[25]. The possibility of differences in the production of $C_3$

phytoliths and $C_4$ phytoliths, which could potentially bias the estimates, has been excluded[11]. Thus, it appears that phytolith analysis is a robust approach for the reconstruction of past $C_4$ biomass.

There are several possible explanations for the discrepancy between phytoliths and soil carbonate $\delta^{13}C$ records. The first is that some studies[6,22] interpret the $\delta^{13}C_{SC}$ values of approximately −8 to −9‰ in the Bahe Formation as indicative of pure $C_3$ vegetation under high water stress. However, our phytolith data suggest that this area was a grass-dominated ecosystem, specifically a $C_3/C_4$ mixed grassland under a relatively humid climate. In addition to the atmospheric $CO_2$ level and $C_4$ vegetation, the $\delta^{13}C$ of soil carbonate is also sensitive to the soil respiration flux[26], which is affected by primary productivity and MAP[27]. Thus, in the Lantian Formation (Red Clay), the $\delta^{13}C_{SC}$ values most likely responded to precipitation and productivity variations rather than to the $C_3/C_4$ composition, as indicated by our phytolith and pollen data. The northward increase in $\delta^{13}C_{SC}$ values suggested by Passey et al.[6] could also reflect the northward decrease in

precipitation and productivity, which resulted in a decreased soil respiration flux. The effects of these climatic gradients on $\delta^{13}C_{SC}$ values in Asia are well documented and may date back to the early Miocene[28]. The second explanation is that phytoliths overestimate the signal of grasses in some ecosystems[29], which could lead to an underestimation of $C_3$ vegetation, such as trees, shrubs and forbs (e.g., Compositae, *Artemisia*). Pollen data from the Chinese Loess Plateau suggest that the percentage of trees was ~15% during the late Miocene[30,31] (Fig. 4b); this percentage largely decreased at ~4 Ma, which is consistent with the positive shift in $\delta^{13}C_{SC}$ values, leading to the speculation that the $\delta^{13}C_{SC}$ values most likely reflect the change in precipitation and productivity rather than a $C_3/C_4$ shift (Fig. 4d).

Our phytolith assemblages-derived temperature and precipitation (percentage of Pooideae grasses) exhibit minor increasing trends starting ~7.0 Ma, with a marked decrease at ~4.2 Ma (Fig. 4a). These relatively warm and humid periods during the late Miocene to early Pliocene, as well as late Pliocene cooling and drying, were captured by various evidence in East Asia, such as pollen data from Linxia[32], Chaona[33,34], Xifeng[30] and Baode[31] (Fig. 4b). An integrated study of the Neogene pollen record also revealed a climate shift to relatively warm and humid conditions during the early Pliocene, as suggested by the resumption of woodland vegetation in China[16]. The shift to relatively warm and humid conditions at ~7.0 Ma is also suggested by fossil mammal data from the same profile as that in this study, which is characterized by the presence of species that lived in a closed canopy[17]. The presence of extremely dry-adapted *Paracamelus* and the marked decrease in species abundance (Fig. 4c) suggest a shift to a dry and cold climate in the late Pliocene across large area of East Asia; the reconstructed precipitation based on the integrated study of hypsodonty also suggests a relatively humid climate during 8~5 Ma, subsequently becoming drier[35]. Our data suggest an even colder and drier Quaternary compared with the late Pliocene, although the uncertainty is large due to the sparse data during this interval (Figs. 3, 4). However, climate change from the late Pliocene to Quaternary has been well illustrated by pollen[36], mammal fossils[37] and sedimentary change[38], which support our results.

Pollen-based quantitative reconstruction of the palaeoclimate in northern China suggests that the MAT and MAP during the late Miocene were higher than those at present[39]. Our phytolith-based quantitative reconstruction in the Weihe Basin also suggests higher MAT and MAP as well as higher monsoon precipitation during the late Miocene to early Pliocene (MAT range of 11~15.3 °C; MAP range of 812~1673 mm; the WMMP range of 76~199 mm; the WSP range of 427~1197 mm) than at present (MAT = 13.8 °C, MAP = 569 mm; the WMMP = 95 mm; the WSP = 439 mm), among which the WMMP and the WSP are direct indicators of the monsoon precipitation. Pollen-based quantitative reconstruction during the late Pliocene is very rare in northern China, except for one study in Yushe Basin with median values of MAT = 11.8 °C and MAP = 948 mm[40], which were lower than those at all late Miocene sites in the eastern region of northern China[39] (four sites, median values of MAT ranging from 12.9 to ~15.9 °C, MAP ranging from 971 to ~1169 mm) and higher than the present values (MAT = 8.8 °C, MAP = 579 mm).

All these data confirm that a large area of East Asia was affected by a major climatic shift in the late Pliocene. The decrease in temperature was probably influenced by global cooling, while the decrease in precipitation (MAP, the WMMP and the WSP) was probably influenced by a decrease in Asian monsoon rainfall because most of the precipitation in East Asia is caused by the monsoon circulation and the history of the Asian monsoon could be dated back to Miocene[41,42] or Eocene[43]. Several possible drivers have been proposed to explain Asian

climate change since the late Miocene, such as the growth of the Tibetan Plateau[2] and global cooling[38]. A decrease in atmospheric $CO_2$ levels during the late Pliocene has also been suggested as the trigger for Northern Hemisphere glaciation[44] as well as the East Asia monsoon climate[45]. However, $pCO_2$ had been below the threshold for Northern Hemisphere glaciation[46] since before the Pliocene (Fig. 4h). Modelling results suggest that feedbacks related to the growth of ice sheets, sea ice and snow albedo are needed to explain both late Pliocene cooling[47] and the East Asian monsoon climate change since the late Miocene. We therefore view our vegetation shift as strong support for the hypothesis that global cooling has been the main driver of the East Asian monsoon climate change since the late Miocene[38,48]. The percentage of Pooideae grasses used as a qualitative proxy of monsoon climate is strongly correlated with global temperature/ice volume[49] ($R = 0.68$, significant at the 0.01 level). The monsoon precipitation belt is associated with the location of the intertropical convergence zone (ITCZ) and global cooling controls monsoon precipitation by pushing the ITCZ towards the tropical region, resulting in a change in the location of the monsoon precipitation belt and a reduction in the monsoon precipitation in East Asia[50,51]. Inconsistencies also exist between global cooling and the East Asian monsoon climate; our phytoliths and other records (Fig. 4a–c) suggest a relatively warm and humid period during the late Miocene to early Pliocene in East Asia, which is inconsistent with the late Miocene cooling suggested by the global sea surface temperature record[52] (Fig. 4f). Other triggers or driving mechanisms are thus required to explain this inconsistency.

In summary, our study provides a new direct record of vegetation since the late Miocene. The vegetation shifts in the late Pliocene revealed marked decreases in temperature and precipitation, which coincide with other evidence of cooling and a decrease in monsoon rainfall in East Asia. We suggest that this decrease in Asian monsoon rainfall was most likely driven by global cooling, which shifted the location of the ITCZ and associated monsoon precipitation belt.

## Methods

**Field work and geological framework**. The study site at Lantian in the Weihe Basin is situated on the southeastern margin of the Chinese Loess Plateau, close to the northern foothills of the Qinling Mountains (Fig. 1). Over 7000 m of lacustrine-fluvial and aeolian sediments have been deposited in the Lantian area since the Eocene, and the basin can be regarded as an Asian monsoon rain gauge since ~50 Ma. The sedimentary sequence includes the Honghe, Bailuyuan, Lengshuigou, Koujiacun, Bahe and Lantian Formations, and is capped by the Quaternary loess-palaeosol sequence. The deposits have been investigated since the 1950s and their lithostratigraphy and biostratigraphy are established[53]. In this study, we focus on the late Neogene sediments of the Bahe Formation (fluvial-lacustrine deposits), the Lantian Formation (aeolian Red Clay) and the Quaternary loess-palaeosol sequence. The chronology of these sequences is well established (Supplementary Fig. 2). A total of 133 samples from the sequence were selected for phytolith analysis; 87 samples contain phytoliths and the results are presented herein (Supplementary Table 2). In the Quaternary loess-palaeosol sequence, our data span the last interglacial-glacial cycle (~130 ka); the older loess-palaeosol samples did not yield enough phytoliths, which may have been destroyed by weathering, as indicated by phytolith images viewed under the microscope (Supplementary Fig. 1). Phytoliths are relatively abundant in the aeolian Red Clay deposits (2.6~7 Ma), with phytoliths preserved in both soil accumulation horizons and carbonate nodule horizons (Supplementary Fig. 2). Seven horizons yielded abundant phytoliths in the Bahe Formation (7~11 Ma), a fluvio-lacustrine deposit consisting of brown siltstone or brown mudstone with indications of palaeosol formation (e.g., root traces and carbonate nodules); in this case, the phytoliths may have originated either from the regional vegetation and were transported by rivers or from the in situ vegetation.

**Phytolith extraction and classification**. Phytoliths were extracted from sediments using a slightly modified version of the Piperno procedure[12]. The procedure consists of deflocculation with sodium pyrophosphate ($Na_4P_2O_7$) and then treatment with 30% hydrogen peroxide ($H_2O_2$) and cold 15% hydrochloric acid (HCl), followed by heavy liquid separation using zinc bromide ($ZnBr_2$, density 2.35 g/cm³) and mounting on a microscope slide with Canada balsam. From each extracted

sample, at least one microscope slide was prepared for phytolith counts and analysis using a compound microscope at ×400 magnification. The preservation status of each assemblage was determined using the semi-quantitative scheme of Strömberg and McInerney[11]. In the case of 38 samples, phytolith preservation ranged from fair to very good and these data were selected for the quantitative palaeoclimatic reconstruction (Supplementary Table 2). Phytoliths were classified following Strömberg and McInerney[11]. Diagnostic phytoliths were assigned to the following classes: FI TOT, CH TOT, POOID-D, POOID-ND, PAN, CHLOR, PACMAD general, OTHG, AQ, NDG and NDO (see Supplementary Table 1 and Supplementary Fig. 1 for details of the classification). NDG and NDO are considered non-diagnostic and were therefore excluded from the vegetation analysis. As differences in regional vegetation result in differences in the dominant phytolith morphotypes, we made one modification to the classification according to the native vegetation; details are presented in the Supplementary Note 1.

**Analysis of vegetation and habitat type.** The analytical approaches in this part follow Strömberg and McInerney[11]. Vegetation structure is inferred by comparing forest indicator phytoliths (FI TOT) and diagnostic grass phytoliths (GSSC). To examine spatiotemporal changes in tree cover, we used the ratio of the sum of forest indicator phytoliths to the sum of forest indicator phytoliths and GSSCs, i.e., FI TOT/ (FI TOT + GSSC), and confidence intervals (95%, unconditional case, using the total count as the sample size) were calculated for the FI-t ratio (Supplementary Table 2). We mainly consider broad patterns of tree cover, focusing on relative changes in plant communities. We inferred the position in the landscape (microhabitat) and proximity of the sites to water by using the frequency of phytoliths typical of wetland plants (AQ; Supplementary Table 1), semi-quantitative estimations of the relative abundances of sponge spicules and available sedimentological information.

We analysed the grass community composition in terms of open-habitat vs. closed-habitat grasses (OH TOT vs. CH TOT) and potential $C_4$ vs. $C_3$ grasses. For the latter case, we compared GSSCs that are likely produced by grasses belonging to the PACMAD clade (which are predominantly $C_4$, based on modern vegetation data) with GSSCs that are produced by grasses belonging to the BOP clade (Bambusoideae–Oryzoideae–Pooideae), which are purely $C_3$, by comparison with modern vegetation data[9]. The relative abundance of $C_4$ grasses in the entire vegetation was similarly inferred as the relative abundance of $C_4$ grass phytoliths vs. all other diagnostic phytoliths. For quantitative reconstruction of $C_4$ biomass, we examined the relative abundance of panicoid + chloridoid GSSCs (PAN + CHLOR), which provided a rough minimum estimate of $C_4$ biomass, and examined the relative abundance of all PACMAD GSSCs (PACMAD TOT), which provided a rough maximum estimate of $C_4$ biomass:

$$C_{4min}(\%) = (PAN + CHLOR)/(GSSC - OTHG) \times (GSSC)/(FI\ TOT + GSSC)\%$$
(1)

$$C_{4max}(\%) = (PACMAD\ TOT)/(GSSC - OTHG) \times (GSSC)/(FI\ TOT + GSSC)\%$$
(2)

We excluded non-diagnostic, unknown and unidentifiable GSSCs (OTHG in Supplementary Table 1) from these calculations. Most of the OTHGs were broken or otherwise obscured GSSCs that could not be identified. Some phytolith morphotypes from $C_3$ grasses (bilobates and crenates) and $C_4$ grasses (bilobates and polylobates) are difficult to identify if they are broken or partially obscured on the microscope slide; this type of phytolith was classified as OTHGs to increase the accuracy of the $C_4$ biomass estimation.

**Quantitative reconstruction.** The coexistence approach uses the climate tolerances of all the nearest living relatives known for a given fossil flora and has been widely used to reconstruct palaeoclimate with pollen and plant macrofossils[54]. Here we use the principle of coexistence approach to phytoliths to distinguish many subfamilies of Poaceae[12]. The climate range of each subfamily of Poaceae is fundamental for using phytoliths to reconstruct past climate quantitatively through the coexistence approach and reconstructing the regional climate requires knowledge of the climate range of regional species. This range is calculated from the combination of modern distribution data and the climatic data of each species, and building a database containing such information is therefore required (Supplementary Tables 3, 4). Theoretically, if there are enough detailed data from the modern ecosystem, then the modern analogue matching procedure (modern analogue method, MAM) can be applied. Unfortunately, these data are not available for open access. With limited ecosystem data, we assessed the dominant subfamily from the fossil phytolith assemblage (assuming it is a grass-dominated ecosystem) and matched it with a modern ecosystem that is dominated by this subfamily. We further checked the optimum climate value of these dominant species, along with their distribution, to attempt to narrow the climate range reconstructed by the coexistence approach (Supplementary Fig. 4 and Supplementary Table 5).

To build a regional (China) species-climate database, we first built a new global species-climate database (Supplementary Note 2, global species-climate dataset), which consisted of a total of 9,390,524 collections spanning 567 genera and 6010 species with corresponding MAT, the warmest month mean temperature, the coldest month mean temperature, MAP, the WMMP, the coldest month mean precipitation, the WSP (April to September) and difference in temperature of the warmest and coldest months. For regional species, we extracted all Poaceae species records in the Flora of China (http://frps.iplant.cn/), which consisted of 247 genera and 1960 species. After matching with the global species and purging invasive species or introduced species, the dataset of China included 177 genera and 598 species with corresponding climate parameters. The classification follows the most recent phylogenetic studies[55,56]. For ecosystem data, we extracted all ecosystem types and their distributions in China (http://www.ecosystem.csdb.cn); the database consists of 866 types, including 586 formations and 280 sub-formations in which 105 ecosystem types are dominated by Poaceae species. After matching with climate data, the dataset consisted of 64 Poaceae species with corresponding distribution and climate parameters (Supplementary Fig. 4b, represented by the red dot).

## Data availability
All relevant data that support the findings of this research are available from the corresponding author upon request.

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

## Acknowledgements

We thank Professors Zhengtang Guo, Zhaoqun Zhang, Daniel O. Breecker, Shejiang Wang, Timothy D. Herbert, Anu Kaakinen, Jonathan Adams and colleagues from US NSF Dust-PIRE project for their valuable assistance. We appreciate Hanzhi Zhang, Wenfeng Sun, Yao Wang, Genghong Wu, Shuyue Li, Pinxin Jiang, Mengyao Jiang and Shuangwen Yi for help in the field and laboratory. This research was supported by the Ministry of Science and Technology of China (the Global Change Program, grant number 2016YFA0600503) and the National Natural Science Foundation of China (grant numbers 41690111 and 41888101).

## Author contributions

H.L. designed and organized this study. H.L., H.W., L.Z., H.Z., F.L. and Y.W. performed the field work. H.W. carried out the laboratory analysis. H.W. and H.L. wrote the manuscript and all the authors discussed and contributed to the manuscript.

## Competing interests

The authors declare no competing interests.
