## [Peer Review File · Nature Communications]

Reviewers' comments:

Reviewer #1 (Remarks to the Author):

This paper provides the first phytolith-based record of vegetation change in eastern Eurasia (specifically China) during the late Miocene-Pleistocene. The data presented by the Wang et al. convincingly demonstrates a shift from C4 (or PACMAD) dominated grasslands to pooid-dominated grasslands at ~4 Ma, counter to much of the published isotopic results (both tooth enamel and soil carbonate-based). These results should be of interest to a wide audience of biologists and geologists.

The paper is well written and researched and, as I indicated, very appropriate for Nature Communications. Thus, I only have fairly minor comments, and recommend this paper for publication pending revisions. I have made edits and remarks throughout the ms and supplementary materials. Here are some of the more substantial points:

1. The authors use a China-specific database to extract climatic envelopes for their grass subfamilies. What they get seems somewhat at odds with Edwards and Smith's (2010) analysis of global grass distributions. I would like them to discuss these differences in more detail and what the implications of them are for the interpretation of their data. For example, if Chinese chloridoids today are more wet-adapted than grasses globally, can we reliably use this climate envelope for the late Miocene?

2. It seems to me that this new record is not just at odds with the soil carbonate-based carbon isotopic record but also the carbon isotope records from tooth enamel (e.g., Passey et al. 2009). I would like the authors discuss this in more detail. I would also see them back up the claim that the soil record is mainly showing changes in precipitation and productivity instead of C3-C4 differences. This conclusion implies that most or much of the soil carbon isotope record is meaningless from a C3-C4 standpoint. It may well be that there's a lot of data pointing in this direction (I have seen it mentioned, but never causing an outright rejection of conclusions based on these types of data), but they need to be more thoroughly discussed.

3. I would love to see a discussion of if there were any trends in which samples yielded phytoliths vs. those that did not. This information would be very helpful for phytolith researchers such as myself. For example, I have previously extracted hundreds of samples from China and Mongolia, and have only gotten phytoliths from diatomaceous lake sediments, never from the red clays etc. (see discussion about preservation in Strömberg et al., 2018). Thus, it would be good to know if there are particular facies that might be more likely to yield biosilica.

Caroline Strömberg

Reviewer #2 (Remarks to the Author):

Comments for authors (rev. Torsten Utescher):

This excellent paper presents a first comprehensive record of phytoliths from a sedimentary sequence in Central China covering the late Miocene to Pleistocene while time control is provided by magnetostratigraphy. Phytoliths allow for a direct access of local vegetation signals, therefore, this record is of high importance and interest to a broad readership. The evaluation of the phytolith data greatly profits from the application of various, partly novel methodological approaches to quantify palaeovegetation and climate. The discussion of the results in the context of data from other proxies is very well done. Moreover, extensive supplementary materials are provided.

I recommend publication of this manuscript but ask the authors to consider a couple of points detailed below. I hope my comments are helpful.

Comments

1)

Title of the manuscript:

As already evident from the abstract and sampled record, the main issue is a Phytolith record since the late Miocene (~11 Ma) and its palaeovegetational and climatic implications while data on the monsoon system and its intensity cannot be assessed directly but are inferred from other observations (composition of the local vegetation, estimate of precipitation rates). For that reason I think that the manuscript title is somewhat misleading and should be modified.

2)

Line 218: avoid term "amelioration" in the climatic context. I suggest to talk about a shift to warmer/more humid conditions..

3)

Lines 261 and following, Figure 4:

You state that "phytoliths... suggest a relatively warm and humid period during the late Miocene to early Pliocene in East Asia, which is inconsistent with late Miocene cooling suggested by the global sea surface temperature (SST) record. I think that the late Miocene cooling has to be understood as a process connected to steepening of the pole-to-equator temperature gradient that already set on at the Mid-Miocene Transition and culminated with ephemeral Northern Hemisphere glaciations between 6.0 and 5.5 Ma. This is also evident from both SST and benthic records you show. Therefore I think that the grey highlighted interval you show in fig. 4 is not so useful.

Moreover it should be stressed here that your archive reveals regional data that are not necessarily congruent with the global SST trend and the benthic stack. We now also very warm and humid phases in the late Miocene and Pliocene (early and late) from European continental records (cf. e.g. Mosbrugger et al., 2005, and many others). Since you have only few rich samples available for the late Pliocene it has to be considered that your record might not resolve warm and humid phases in that time-span, given the high variability of climate at that time..

4)

Supplement, Figure S5

I think that the position of the green line, bordering the distribution of forest vegetation should be removed or at least dashed because there are no data points supporting its course. By the way: you characterize the late Miocene as a warmer and more humid phase in the study area compared to present-day. So why did you put the forest border further to the South?

5)

Supplement 2. Global species-climate dataset

This section needs a linguistic check-up and contains various typos.

Reviewer #3 (Remarks to the Author):

This paper investigated phytoliths (plant silica) from sediments spanning the last 11 million years (Ma) in Central China in order to infer "monsoon rainfall variation during the late Pliocene", the period from 3.6 to 2.56 Ma.

The study reports a grass flora dominated by members of the PACMAD clade of Poaceae during 11 to 4 Ma, while members of Pooideae dominate younger floras.

(Note that the observed shift in dominance of grass clades appeared in the early Pliocene, not in the late Pliocene as indicated in the title and throughout the text.)

My main observations when reading this paper are as follows:

1. Of 50 references cited in this work, only a single is from after 2015. As knowledge about the

evolution of the Asian Monsoon increases rapidly, this is not trivial. See my detailed comments below.

2. Changes in Mean annual precipitation (MAP) are used to reconstruct monsoon intensity. However, MAP is not suitable to infer monsoon intensity. Instead it is precipitation seasonality that should be used when reconstructing fluctuations in East Asian monsoon.

Detailed comments:

Title: Change to "... during the Pliocene"

Line 10: First and second part of this sentence are not logically connected. The first is a statement about AM during a short period during "late" Pliocene. However, AM existed since the Eocene.

Line 15: See Dong et al. 2018 (Journal of Asian Earth Sciences 158) for a mid-Miocene C4 expansion on the Chinese Loess Plateau.

Line 16: 4 Ma would be early Pliocene (see Cohen et al., 2013 updated. version 2018/07)

Line 20: Ok, maybe true but it is hardly possible to express this by MAP values, which do not tell anything about precipitation seasonality. It is precipitation seasonality that distinguishes weak and strong monsoon not MAP.

Line 22: This is not novel.

Line 32: Awkward phrasing. An argument cannot come from the Arabian Sea. Please, re-phrase.

Lines 33/34: This would mean development of a Mediterranean climate, which is the exact opposite of a monsoon climate.

I don't get your point here. Is this meant to be a contradiction to what is stated in the sentence before?

Line 39: See a more recent study, Dong et al. 2018 (Journal of Asian Earth Sciences 158), for evidence of a C4 expansion on the CLP during the middle Miocene.

Line 42: You state here that the environmental niches of C4 grasses are still unclear. However, how, then, can you use C4 grasses to reconstruct palaeoenvironment/climate?

Also, I wonder whether there has not been any progress on this topic since 2015? See, for example, Reich et al 2018 (Science) and comments to this paper, and many others.

Line 59: Delete ", or the reverse"

Line 88: Again, I wonder what the novelty of this finding is.

See, for example, Jiang & Ding, 2008 (Palaeogeography, Palaeoclimatology, Palaeoecology 265, 30-38), for a comprehensive pollen record for the past 20 Ma for this region.

Lines 102/103: "For the seasonality of precipitation, all subfamilies are similar and indistinguishable".

See my comment above (Abstract). I do not think that MAP values are suitable to distinguish monsoon and non-monsoon climates, but that information on ratios between, for example, 3 consecutive driest months to three consecutive wettest months is needed to quantify monsoon.

Lines 116/117: Be careful with this threshold. MAP <1500mm does not tell anything, this can be a tropical rainforest in Colombia, a laurel forest in Pensacola, a temperate broadleaf forest in Torino....

Line 137: It is unclear to me (also after having read the Methods section) what exactly you did when undertaking a "Coexistence Approach"??

Lines 141-149: Given that you use major clades in Poaceae as nearest living relatives, I do not think that 800-1673 mm or 441-900 mm is very meaningful. I suggest re-phrasing to ca. 800 to 1700 mm and ca. 440 to 900 mm.

Lines 152-53: "Our results reveal that C4 grasses were very abundant in the study area ~11.0 Ma, which is most likely the earliest fossil record of C4 grassland in East Asia"
Not if you include literature from after 2015. See for example Dong et al. 2018 (Journal of Asian Earth Sciences 158).

Line 154: Shanwang has been dated as early Miocene (Yu et al. 2017, Acta Geologica Sinica 91/4).

Line 170: change "Late Miocene" to late Miocene

Line 181: "A controversial issue is that our phytolith data provide a different estimation of C4 grasses compared to that derived from $\delta^{13}\text{C}$ data of soil carbonates"
Again, all this has to be discussed in view of an updated literature database. MANY papers relevant to this have been published since 2015. The controversy may not exist anymore.

Lines 213-266: I wonder what is novel in this chapter. It is a mere compilation of what has been published. That globally averaged isotopic data from benthic foraminifera do not always reflect regional climate is trivial.

References: As stated above, for this topic, it is of utmost importance to keep track with what has been published in most recent times. With a single reference younger than 2015 this is not the case here.

Some journal abbreviations look odd, e.g. P. Natl. Acad. Sci. Usa., Annu. Rev. Ecol. S., Earth Planet. Sc. Lett., Ann. Bot.-London etc etc

Methods:

Lines 516 ff: This chapter needs some clarification. Which were the NLR taxa used for the CA? I suggest a Table could help with NLR taxa and climate intervals indicated for the selected climate parameters.

Line 521: Change "quantitative reconstruct" to reconstruct quantitatively

Line 522: This is valid only if you assume that the same species thrived there since 11 Ma. Excluding any kind of niche evolution in grass clades.

Line 524: Please explain, what you consider a nearest living relative. Did you construct a coexistence interval from all members of a subfamily in a region?

Lines 525 to 527: What does this mean: Do you have these data but you are not allowed to share them with the scientific community?
Or, do these data exist, but you have not seen them?

Supplementary Information

Table S2: Change Plam to Palm

Reviewer #1

1) The authors use a China-specific database to extract climatic envelopes for their grass subfamilies. What they get seems somewhat at odds with Edwards and Smith's (2010) analysis of global grass distributions. I would like them to discuss these differences in more detail and what the implications of them are for the interpretation of their data. For example, if Chinese chloridooids today are more wet-adapted than grasses globally, can we reliably use this climate envelope for the late Miocene?

Response: Thank you for this suggestion. Compared to the data from global grasses (we updated the data from global grasses by following the method of Edwards and Smith, 2010, see Methods), grasses in China show few differences in their temperature preferences: Pooideae still occupy the cool end of the temperature spectrum, and other subfamilies are indistinguishable. However, the precipitation preferences of the grasses are quite different: Pooideae stand apart as inhabiting the driest environments of all subfamilies. Chloridoideae are adapted to more humid environments than Pooideae but drier environments than Panicoideae, Oryzoideae and Bambusoideae (Fig. 2).

In detail, the mean MAT and MAP of Pooideae grasses lived in are 6.1°C and 707 mm with high differentiation, and some species lived at high temperatures (*Alopecurus japonicus*, MAT=15.3°C) and in areas with high precipitation (*Festuca japonica*, MAP=1674 mm). The mean MAT and MAP of Panicoideae grasses are 19.4°C and 1424 mm, respectively, with high differentiation. Some species live in cold and dry environments (*Pennisetum lanatum*, MAT=3.3°C, MAP=504 mm). Chloridoideae is a special subfamily because two species (*Orinus thoroldii*, MAT=0.1°C, MAP=273 mm; *Cleistogenes squarrosa*, MAT=1.5°C, MAP=338 mm) lived in the cold and dry environments that are common in northern China, while the other species lived in the warm and humid environments that are common in southern China (see Fig. S4). Bambusoideae and Oryzoideae tend to live in close habitats that offer good lower limits of temperature and precipitation, *Fargesia spathacea*

(MAT=7.9°C, MAP=738 mm) adapted to the coldest and driest environments in Bambusoideae and *Leersia oryzoides* (MAT=9.7°C, MAP=812 mm) adapted to the coldest and driest environments in Oryzoideae. The detailed climate ranges of each subfamily in China/globally are also shown in Table S3 and Table S4.

For your first question, yes, Chloridoideae grasses in China indeed adapt to more humid environments than Chloridoideae globally except for two species. We suggest that the general “impression” of dry-adapted Chloridoideae was mainly derived from North America, where this subfamily dominated the grassland owing to their short-stature and drought-resistant characteristics (Edward et al., 2010, Science). However, the scenario was different in China: of the 59 species of Chloridoideae recorded in the Flora of China (<http://foc.eflora.cn/>), only 13 of them could be found in America, while most of them (46 species) could be found in Eurasia, Oceania and Africa.

For your second question, we should be careful when using this climate envelope. The assumption we made is that the environmental niches of NLRs are the same as fossil species on a regional scale, which means that there is no mass migration of species across the continent (e.g., from North America to Asia) while ignoring the possible changes in environmental niches affected by the evolution process. We added a sentence to discuss these problems following your suggestions (please see: page 6, lines 120~126).

2) It seems to me that this new record is not just at odds with the soil carbonate-based carbon isotopic record but also the carbon isotope records from tooth enamel (e.g., Passey et al. 2009). I would like the authors discuss this in more detail. I would also see them back up the claim that the soil record is mainly showing changes in precipitation and productivity instead of C3-C4 differences. This conclusion implies that most or much of the soil carbon isotope record is meaningless from a C3-C4 standpoint. It may well be that there's a lot of data pointing in this direction (I have seen it mentioned, but never causing an outright rejection of conclusions based on

these types of data), but they need to be more thoroughly discussed.

Response: Thank you. We did not see our data as the opposite of $\delta^{13}\text{C}_{\text{enamel}}$ data. Passey et al. (2009) integrated all $\delta^{13}\text{C}_{\text{enamel}}$ data from different sites into one figure and concluded that C4 expansion in China was not significantly delayed compared to the global C4 event. However, looking into the data of Passey et al. (2009), we found that the $\delta^{13}\text{C}$ of tooth enamel of each site did not exhibit an increasing trend since the late Miocene. For Baode and Yushe sites, there are no $\delta^{13}\text{C}_{\text{enamel}}$ data older than ~7 Ma, and Lantian did not have $\delta^{13}\text{C}_{\text{enamel}}$ data younger than ~7 Ma. The suggested expansion of C4 grasses at ~6 Ma could be the result of sparse data. Hence, we integrated all published $\delta^{13}\text{C}$ data of tooth enamel, sorted the data by location, and found that in Lingtai, Inner Mongolia and Qaidam Basin, the $\delta^{13}\text{C}$ decreased since 4~5 Ma and interpreted these changes as the decreasing trend in C4 grasses (Fig. S5). We are also aware that these sparse data are insufficient to reject the late Miocene expansion of C4 grasses suggested by Passey et al. (2009). We suggest that C4 grasses are very sensitive to regional climate change (the zone of C4 grasses could move ~300 km on the glacial-interglacial scale (Yang et al., 2015, PNAS)). With more data available, especially continuous data from one site, and the puzzle of C4 grasses could be resolved.

[redacted]

On the other hand, the strong East Asian summer monsoons suggested by *Passey et al. (2009)* are consistent with our results. *Passey et al. (2009)* drew their conclusion mainly based on the temporal and spatial distributions of $\delta^{13}\text{C}_{\text{SC}}$ ($\delta^{13}\text{C}$ of soil carbonate) in CLP, and the problem occurred when interpreting the $\delta^{13}\text{C}$ data as a vegetation record. According to the vegetation distribution model provided by *Passey et al. (2009)*, the vegetation in the Baode area should change from desert-steppe (C3 +C4) to desert (C3), and $\delta^{13}\text{C}_{\text{SC}}$ should be more negative during 4~2.6 Ma. However, the data provided by *Suarez et al. (2011)* suggest a positive shift in $\delta^{13}\text{C}_{\text{SC}}$. *Suarez et al. (2011)* interpreted these data as northward-increasing C4 vegetation. New data from the more northern Erlian Basin suggested more positive $\delta^{13}\text{C}_{\text{SC}}$ values during 14~5 Ma (mean value of -6.0‰ compared to mean value of -6.4‰ in Baode (*Caves et al., 2016, Geology*)). *Caves et al. (2016)* integrated a large amount of soil carbonate data (2,236 data records) in Asia since the Miocene and interpreted the changes in $\delta^{13}\text{C}_{\text{SC}}$ as changes in both primary productivity and mean annual precipitation. We thought that interpreting new data as northward-increasing C4

vegetation means more abundant C4 in northern central Asia, which is unlikely. Hence, we support the results of *Caves et al.* (2016), who interpreted the changes in $\delta^{13}\text{C}_{\text{SC}}$ as changes of both primary productivity and mean annual precipitation, which could resolve the conflicts between our data and the $\delta^{13}\text{C}$ of soil carbonate.

We do not outright reject the use of $\delta^{13}\text{C}_{\text{SC}}$ to interpret C3/C4 vegetation because there are many factors that could affect $\delta^{13}\text{C}_{\text{SC}}$, such as $p\text{CO}_2$, C3/C4, primary productivity and precipitation, and it is difficult to distinguish one from another in many sites. However, in some regions, the $\delta^{13}\text{C}$ of soil carbonate is useful for interpreting C3/C4 vegetation (*Quade et al.*, 1989, Siwalik, northern Pakistan). This study suggests that we should be careful when interpreting $\delta^{13}\text{C}$ data of soil carbonate from East Asia since the Miocene. For example, $\delta^{13}\text{C}_{\text{SC}}$ values of approximately -8~-9 ‰ could be pure C3 vegetation under conditions of high water stress or C3/C4 mixed grassland under a relatively humid climate. Usually, other proxies such as pollen or phytolith data would help to interpret $\delta^{13}\text{C}_{\text{SC}}$ data (see Discussion, page 10, lines 210~230).

3) I would love to see a discussion of if there were any trends in which samples yielded phytoliths vs. those that did not. This information would be very helpful for phytolith researchers such as myself. For example, I have previously extracted hundreds of samples from China and Mongolia, and have only gotten phytoliths from diatomaceous lake sediments, never from the red clays etc. (see discussion about preservation in *Strömberg et al.*, 2018). Thus, it would be good to know if there are particular facies that might be more likely to yield biosilica.

Response: The preservation of phytoliths has always been a problem for phytolith analysis. We briefly discuss the preservation in the text (page 4, lines 79~90), and we discuss this in more detail here because we performed extensive extra work to explore this problem. We have examined the preservation of phytoliths in three types of sediments: loess, 'Red Clay' and fluvio-lacustrine deposits.

In the loess deposit, samples older than the last interglacial period (~130 ka) yielded no phytoliths at the Lantian site (34.20°N, 109.24°E). A study of phytoliths in Weinan (34.45°N, 109.53°E) (close to Lantian, Lü et al., 2007, QSR) showed similar results. We tried to extract phytoliths from loess deposits from a northern site, Yulin (38.34°N, 109.74°E) and found that phytoliths were preserved in loess samples only younger than ~70 ka (unpublished data). Taken together, these results indicate that there are three reasons for preservation of phytoliths in loess: 1) loess are porous sediments with high pH values (~8.5, e.g., Liu, 1985), which results in the generally poor preservation of phytoliths; 2) a dry and cold climate in the north produces more aeolian sand, which likely etched/destroyed the phytoliths; 3) a dry and cold climate results in fewer trees and grasses, which leads to low phytolith yields. A combination of all these factors is also possible.

Red Clay deposit yielded relatively abundant phytoliths (13 of 63 samples); however, the concentration of phytoliths ranges from 20~340 grains/g, with an average of 150 grains/g in the 13 well-preserved phytolith samples, which required extensive time for searching and counting. There was no preference for the horizons that were likely to yield more phytoliths, and both soil B horizons and carbonate nodule horizons yielded phytoliths. We also tried to extract phytoliths from other Red Clay sequences from Lingtai (0 of 30 samples) (35.12°N, 107.25°E), Shilou (0 of 30 samples) (36.98°N, 110.81°E), Jiaxian (0 of 30 samples) (38.26°N, 110.00°E) and Xining Basin (0 of 30 samples) (36.66 °N, 101.80°E), unfortunately, none of these profiles yielded enough phytoliths. Red Clay is aeolian deposit with finer grain sizes than loess; physical weathering is less important in Red Clay, and the intensity of chemical weathering and possible diagenesis in Red Clay are nearly identical. Therefore, we suggest that the primary productivity might be important for the concentrations of phytoliths in Red Clay.

In the fluvio-lacustrine deposit from the Bahe Formation, 7 of the 42 samples yielded enough phytoliths. The concentrations of phytoliths were higher in these

samples than in the Red Clay samples, with an average of 650 grains/g. Meanwhile, the faunal composition of Bahe formation was quite flourishing. The Bahean stage (an Asian Land Mammal Age) was named after this site. The depositional units yield phytoliths usually with indications of palaeosol formation (e.g., root traces and carbonate nodules); however, the parent material (C horizon) of palaeosol is unfixed, some of the soils developed in mudstone, some in siltstone, while other mudstone, siltstone and sandstone yielded no phytoliths. The preservation of phytoliths in this deposition sequence is consistent with the prediction that phytoliths can tolerate highly oxidized environments, such as floodplain sediments (Strömberg 2002; Strömberg et al., 2018).

Summary: We thank Professor Caroline Strömberg very much for all the comments and a thorough review of our manuscript. All specific errors or problems have been revised according to the edits and remarks in the manuscript.

Reviewer #2

1) Title of the manuscript:

As already evident from the abstract and sampled record, the main issue is a Phytolith record since the late Miocene (~11 Ma) and its palaeovegetational and climatic implications while data on the monsoon system and its intensity cannot be assessed directly but are inferred from other observations (composition of the local vegetation, estimate of precipitation rates). For that reason I think that the manuscript title is somewhat misleading and should be modified.

Response: Thank you for the suggestion. We revised the title to “Asian monsoon rainfall variation during the Pliocene forced by global temperature change”. We agree that the phytolith data could only directly indicate vegetation and indirectly indicate monsoon intensity. However, the Asian monsoon circulation is the most important

driver in this region to determine the precipitation and vegetation variations, and therefore the phytolith assemblages can indicate the vegetation and the intensity of Asian monsoon circulation. To understand the monsoon behaviour and its forcing mechanism, in particular in the warm Earth of the Pliocene, is important to understand the environmental changes and in central China and most of Asia.

2) Line 218: avoid term “amelioration” in the climatic context. I suggest to talk about a shift to warmer/more humid conditions.

Response: Thank you. We have revised this sentence to “An integrated study of the Neogene pollen record in China also revealed a climate shift to relatively warm and humid conditions during the early Pliocene, as suggested by the resumption of woodland vegetation.”

3) Lines 261 and following, Figure 4:

You state that “phytoliths... suggest a relatively warm and humid period during the late Miocene to early Pliocene in East Asia, which is inconsistent with late Miocene cooling suggested by the global sea surface temperature (SST) record. I think that the late Miocene cooling has to be understood as a process connected to steepening of the pole-to-equator temperature gradient that already set on at the Mid-Miocene Transition and culminated with ephemeral Northern Hemisphere glaciations between 6.0 and 5.5 Ma. This is also evident from both SST and benthic records you show. Therefore I think that the grey highlighted interval you show in fig. 4 is not so useful.

Moreover it should be stressed here that your archive reveals regional data that are not necessarily congruent with the global SST trend and the benthic stack. We now also very warm and humid phases in the late Miocene and Pliocene (early and late) from European continental records (cf. e.g. Mosbrugger et al., 2005, and many others). Since you have only few rich samples available for the late Pliocene it has to be considered that your record might not resolve warm and humid phases in that

time-span, given the high variability of climate at that time.

Response: Yes, we agree that regional data are not necessarily consistent with global SST. We also agree that we did not obtain enough data to describe this relatively warm and humid period during the late Miocene to early Pliocene. More detailed work (high resolution) is needed to obtain a reliable conclusion. We deleted the grey highlighted interval about late Miocene cooling in Fig. 4 following your suggestion.

4) Supplement, Figure S5

I think that the position of the green line, bordering the distribution of forest vegetation should be removed or at least dashed because there are no data points supporting its course. By the way: you characterize the late Miocene as a warmer and more humid phase in the study area compared to present-day. So why did you put the forest border further to the South?

Response: Thank you for the suggestion. We agree that this green line is mainly based on pollen and fossil mammal data in the reference (Jiang and Ding, 2009; Qiu et al., 2013). We dashed the green line in Figure S5 and moved it a little further north following your suggestion.

5) Supplement 2. Global species-climate dataset

This section needs a linguistic check-up and contains various typos.

Response: We apologize for the linguistic problem. We have checked this section thoroughly and corrected the errors accordingly.

Reviewer #3

1) Of 50 references cited in this work, only a single is from after 2015. As knowledge about the evolution of the Asian Monsoon increases rapidly, this is not trivial. See my detailed comments below.

Response: Thank you for the suggestion. We added some recent published references, such as the following:

Caves, J.K. *et al.* The Neogene de-greening of Central Asia. *Geology* **44**, 887-890 (2016).

Hyland, E.G., Sheldon, N.D., Smith, S.Y. & Strömberg, C.A.E. Late Miocene rise and fall of C4 grasses in the western United States linked to aridification and uplift. *Geol. Soc. Am. Bull.* **131**, 224-234 (2019).

Strömberg, C.A.E., Dunn, R.E., Crifò, C. & Harris, E.B. Phytoliths in Paleoecology: Analytical Considerations, Current Use, and Future Directions. in *Methods in Paleoecology: Reconstructing Cenozoic Terrestrial Environments and Ecological Communities* (eds. Croft, D.A., Su, D.F. & Simpson, S.W.) 235-287 (Springer International Publishing, Cham, 2018).

Lu, H. *et al.* Cenozoic depositional sequence in the Weihe Basin (Central China): A long-term record of Asian monsoon precipitation from the greenhouse to icehouse Earth. *Quaternary Sciences* **38**, 1057-1067 (2018).

Soreng, R.J. *et al.* A worldwide phylogenetic classification of the Poaceae (Gramineae). *J. Syst. Evol.* **53**, 117-137 (2015).

Soreng, R.J. *et al.* A worldwide phylogenetic classification of the Poaceae (Gramineae) II: An update and a comparison of two 2015 classifications. *J. Syst. Evol.* **55**, 259-290 (2017).

2) Changes in Mean annual precipitation (MAP) are used to reconstruct monsoon intensity. However, MAP is not suitable to infer monsoon intensity. Instead it is precipitation seasonality that should be used when reconstructing fluctuations in East Asian monsoon.

Response: We agree that it is more appropriate to use the precipitation seasonality to define strong/weak monsoons. However, we suggest that MAP could still represent the intensity of the East Asian monsoon for several reasons:

1) The monsoon circulation has probably remained similar to the current conditions since the Eocene (e.g., Quan et al., 2012, 3Paleo; Spicer et al., 2016, EPSL; Lu et al., 2018, Quaternary Sciences; Caves and Chamberlain, 2018, Earth-Science Reviews), or later (An et al., 2001, Nature; Guo et al., 2002, Nature; Wang et al., 2017, ESR). A monsoon climate means an increased percentage of warm season precipitation and therefore may make a greater contribution to the MAP. Thus, we suggest that the stronger summer monsoon means a high amount of warm season precipitation, which usually accompanies a higher MAP. The MAP in Weihe Basin could be used as a proxy of the East Asian monsoon strength.

2) We are not the only authors that have used MAP to indicate monsoon rainfall (e.g., Nie et al., 2014, Scientific Reports). Except for a few proxies (e.g., $\delta^{18}\text{O}$ values in ostracods, Bougeois et al., 2018, EPSL; T Δ 47, Suarez et al., 2011, Geology; Page et al., 2019, Geology) and climate simulation results that could provide seasonality information on the tectonic scale, most studies usually use “precipitation/rainfall” or “monsoon precipitation/rainfall” to represent monsoon intensity (e.g., An et al., 2015, Annual Review of Earth Planetary Science; Wang et al., 2017, Earth-Science Review).

3) Title: Change to “... during the Pliocene”

Response: Thank you, this change has been made.

4) Line 10: First and second part of this sentence are not logically connected. The first is a statement about AM during a short period during “late” Pliocene. However, AM existed since the Eocene.

Response: Thank you for this comment. We corrected this sentence to “The Asian monsoon variations under global temperature changes during the Pliocene are still debated, which impedes the scientific knowledge of the forcing mechanism of Asian monsoon variations during this period.”

5) Line 15: See Dong et al. 2018 (Journal of Asian Earth Sciences 158) for a mid-Miocene C4 expansion on the Chinese Loess Plateau.

Response: Thank you for this comment. We were aware of this paper by *Dong et al.* (2018) before submitting our manuscript. However, we still believe our phytoliths provide **direct fossil evidence** for the early C4 expansion. We doubted the interpretation of $\delta^{13}\text{C}$ of soil carbonate by *Dong et al.* because there are many other $\delta^{13}\text{C}_{\text{sc}}$ data that are more positive than -5.0 ‰ in northern central Asia and northern China since the Miocene (Caves et al., 2016, Geology); *Caves et al.* (2016) interpreted $\delta^{13}\text{C}_{\text{sc}}$ data as both primary productivity and mean annual precipitation rather than C4 grasses. We discussed the debates between phytoliths and $\delta^{13}\text{C}_{\text{sc}}$ data in detail in the manuscript.

Please see our response to Reviewer #1 regarding this question.

6) Line 16: 4 Ma would be early Pliocene (see Cohen et al., 2013 updated. version 2018/07)

Response: Thank you, this change has been made.

7) Line 20: Ok, maybe true but it is hardly possible to express this by MAP values, which do not tell anything about precipitation seasonality. It is precipitation seasonality that distinguishes weak and strong monsoon not MAP.

Response: We agree that it is more appropriate to use the precipitation seasonality to define strong/weak monsoons. However, we have several reasons why MAP could represent the monsoon rainfall. Please see our response to your first comment.

8) Line 22: This is not novel.

Response: we agree that global temperature driving hypothesis is not novel, and many studies, including ours, proposed this hypothesis years ago (e.g., Jiang and Ding, 2008, 3Palaeo; Lu et al., 2008, 2010; Lu and Guo, 2014; Lu et al., 2018). In this paper we use this hypothesis to interpret our newly obtained phytolith data.

9) Line 32: Awkward phrasing. An argument cannot come from the Arabian Sea. Please, re-phrase.

Response: Thank you. We corrected this sentence to “However, one of the objections for increased warm-season precipitation comes from the evidence in the western Arabian Sea, in the form of reduced summer precipitation and/or increased winter precipitation that are required to explain the increased δD values of leaf wax.”

10) Lines 33/34: This would mean development of a Mediterranean climate, which is the exact opposite of a monsoon climate.

I don't get your point here. Is this meant to be a contradiction to what is stated in the sentence before?

Response: Thank you for the comment. We apologize for the linguistic problem, and we have corrected this sentence; see our response to your 9th comment.

11) Line 39: See a more recent study, Dong et al. 2018 (Journal of Asian Earth Sciences 158), for evidence of a C4 expansion on the CLP during the middle Miocene.

Response: Thank you for the suggestion. The main point of this paragraph is to discuss the debate regarding the Asia monsoon variations during the Pliocene, and the

reason for this discrepancy is the use of stable carbon isotopic analysis to interpret C4 grasses. Meanwhile, we doubted the interpretation of $\delta^{13}\text{C}$ of soil carbonate by *Dong et al.* (2018); please see our response to your 5th comment.

12) Line 42: You state here that the environmental niches of C4 grasses are still unclear. However, how, then, can you use C4 grasses to reconstruct palaeoenvironment/climate?

Also, I wonder whether there has not been any progress on this topic since 2015?

See, for example, Reich et al 2018 (Science) and comments to this paper, and many others.

Response: Thank you for the comments. For your first question, we did not use the percentage of C4 grasses; instead, we used phytolith assemblages to reconstruct the palaeoclimate. Therefore, the following two problems were resolved by our study: 1) different lineages of C4 grasses have different environmental niches; 2) when discussing C4 expansion, you do not know what kind of C3 expanded, it could be C3 forest, C3 shrubs or C3 grasslands, so you cannot tell which environment favours the growth of C4 grasses. However, the phytolith assemblages could perfectly solve the above two problems (we discussed it in detail in the manuscript).

For your second question, there are plenty of studies that focus on the progress on the difference between C4 and C3 photosynthesis based on contrasting experiments of few selected species (Ripley et al., 2015, Ecology; Taylor et al., 2018, Journal of Experimental Botany; Reich et al., 2018, Science). However, we would like to use these results to interpret geological records, which need integrated study of C4 and C3 grasses worldwide, unfortunately this kind of study is rare (Edward and Smith, 2010, PNAS; Edward et al., 2010, Science).

13) Line 59: Delete “, or the reverse”

Response: Thank you, this phrase has been deleted.

14) Line 88: Again, I wonder what the novelty of this finding is.

See, for example, Jiang & Ding, 2008 (Palaeogeography, Palaeoclimatology, Palaeoecology 265, 30-38), for a comprehensive pollen record for the past 20 Ma for this region.

Response: Our study is a pioneering work in the use of phytoliths as a proxy to study monsoon climate change extending to the late Miocene, which is much longer than the previous investigation. One unique advantage of phytolith analysis is that phytoliths offer direct fossil evidence of C4 grasses and associated climate change. Moreover, we constructed a new grass-climate database, and based on this database, we used phytolith assemblages to quantitatively reconstruct Asian monsoon changes since the late Miocene for the first time.

Jiang and Ding (2008, Palaeogeography, Palaeoclimatology, Palaeoecology 265, 30-38) did a good job reconstructing the palaeoclimate by using the pollen record over the past 20 Ma. However, there are three drawbacks of their study. 1) Their timescale for the depositional sequence was notoriously debated, Wang et al. (2011, Tectonophysics) concluded that the timescale should be revised. 2) The pollen could originate from the Liupan Mountains (Lin et al., 2010, Journal of Asian Earth Sciences; Zheng et al., 2006, EPSL) rather than regional sources; thus, their palaeoclimatic implications should be further investigated. 3) Their record is far from the monsoon core region, in which the westerlies could possibly impact the palaeoclimate changes.

15) Lines 102/103: “For the seasonality of precipitation, all subfamilies are similar and indistinguishable”.

See my comment above (Abstract). I do not think that MAP values are suitable to distinguish monsoon and non-monsoon climates, but that information on ratios

between, for example, 3 consecutive driest months to three consecutive wettest months is needed to quantify monsoon.

Response: We agree that it is more suitable to use the precipitation seasonality to define strong/weak monsoons. However, we have several reasons why MAP could represent the monsoon rainfall; please see our response to your first comment. We calculate the seasonality of precipitation using the CV (coefficient of variation) of the precipitation of twelve months following Edward and Smith, 2010, PNAS.

16) Lines 116/117: Be careful with this threshold. MAP <1500mm does not tell anything, this can be a tropical rainforest in Colombia, a laurel forest in Pensacola, a temperate broadleaf forest in Torino...

Response: Thank you for the suggestion. We deleted this sentence.

17) Line 137: It is unclear to me (also after having read the Methods section) what exactly you did when undertaking a “Coexistence Approach”??

Response: Thank you. We first built a database including the distribution and climate range of each regional species and calculated the climate range of each subfamily using the union (not coexistence) of the climate range of each species belonging to each subfamily. We then used each subfamily of Poaceae as NLRs and applied the principle of CA to phytoliths, which was able to distinguish most subfamilies of Poaceae. Finally, based on the presence/absence of each subfamily, we calculated the climate range of each sample (we discussed it in detail in the Supplementary information, part 2, Global species-climate dataset, Table S3, Table S5).

18) Lines 141-149: Given that you use major clades in Poaceae as nearest living relatives, I do not think that 800-1673 mm or 441-900 mm is very meaningful. I

suggest re-phrasing to ca. 800 to 1700 mm and ca. 440 to 900 mm.

Response: Thank you. We agree that the specific amount of precipitation is not very meaningful. We use original numbers in case readers would like to check this result, as the number changes could cause confusion.

19) Lines 152-53: “Our results reveal that C4 grasses were very abundant in the study area ~11.0 Ma, which is most likely the earliest fossil record of C4 grassland in East Asia”

Not if you include literature from after 2015. See for example Dong et al. 2018 (Journal of Asian Earth Sciences 158).

Response: Thank you for this comment. We were aware of this paper by *Dong et al.* (2018) before submitting our manuscript, which is why we use the term “fossil record”, *Dong et al.* use the stable isotopic composition to reconstruct the C4 grass. Meanwhile, we doubted the interpretation of $\delta^{13}\text{C}$ of soil carbonate by *Dong et al.* (2018); please see our response to your 5th comment.

20) Line 154: Shanwang has been dated as early Miocene (Yu et al. 2017, Acta Geologica Sinica 91/4).

Response: Thank you. We changed “middle Miocene” to “early Miocene”.

21) Line 170: change “Late Miocene” to late Miocene.

Response: Thank you, done.

22) Line 181: “A controversial issue is that our phytolith data provide a different estimation of C4 grasses compared to that derived from $\delta^{13}\text{C}$ data of soil carbonates”

Again, all this has to be discussed in view of an updated literature database. MANY papers relevant to this have been published since 2015. The controversy may not exist

anymore.

Response: We found recent published papers about the comparison between phytolith data and $\delta^{13}\text{C}$ data from soil organic matter ($\delta^{13}\text{C}_{\text{org}}$), such as Cotton et al., 2012, 3Paleo; Chen et al., 2015, 3Paleo; Harris et al., 2017, 3Paleo; Hyland et al., 2019, The Geological Society of America. These studies suggest the general consistency between phytoliths and $\delta^{13}\text{C}_{\text{org}}$ data and inconsistency between phytoliths and $\delta^{13}\text{C}$ data of soil carbonate (e.g., Cotton et al., 2012; Chen et al., 2015).

23) References: As stated above, for this topic, it is of utmost importance to keep track with what has been published in most recent times. With a single reference younger than 2015 this is not the case here.

Response: Thank you. We added some recent published references; please see our response to your first comment.

24) Some journal abbreviations look odd, e.g. P. Natl. Acad. Sci. Usa., Annu. Rev. Ecol. S., Earth Planet. Sc. Lett., Ann. Bot.-London etc etc.

Response: Thank you. These abbreviations have been modified.

25) Methods:

Lines 516 ff: This chapter needs some clarification. Which were the NLR taxa used for the CA? I suggest a Table could help with NLR taxa and climate intervals indicated for the selected climate parameters.

Response: Thank you for the suggestion. We used each subfamily of Poaceae as NLRs, and the climate intervals of each subfamily were calculated from the union (not coexistence) of the climate interval of each species belonging to each subfamily. We added Table S3 in the supplementary information following your suggestion.

26) Line 521: Change “quantitative reconstruct” to reconstruct quantitatively.

Response: Thank you. This has been corrected.

27) Line 522: This is valid only if you assume that the same species thrived there since 11 Ma. Excluding any kind of niche evolution in grass clades.

Response: We agree with you. The assumption we made is that the environmental niches of NLRs are the same as fossil species on a regional scale, which means that there is no mass migration of species across the continent (e.g., from North America to Asia) while ignoring the possible changes in environmental niches affected by the evolution process. We added information on the assumptions and potential sources of error to the manuscript (page 6, lines 120~125).

28) Line 524: Please explain, what you consider a nearest living relative. Did you construct a coexistence interval from all members of a subfamily in a region?

Response: We used each subfamily of Poaceae as NLRs, and the climate intervals of each subfamily were calculated from the union (not coexistence) of the climate interval of each species belonging to each subfamily.

29) Lines 525 to 527: What does this mean: Do you have these data but you are not allowed to share them with the scientific community?

Or, do these data exist, but you have not seen them?

Response: These data exist but are not possible to share due to national security restrictions.

30) Supplementary Information

Table S2: Change Plam to Palm

Response: Thank you. This term has been changed.

References cited

- Lü, H. *et al.* Phytoliths as quantitative indicators for the reconstruction of past environmental conditions in China II: palaeoenvironmental reconstruction in the Loess Plateau. *Quaternary Sci. Rev.* **26**, 759-772 (2007).
- Liu, T. *Loess and environment*, (Science Press, 1985).
- Strömberg, C.A.E. The origin and spread of grass-dominated ecosystems in the late Tertiary of North America: preliminary results concerning the evolution of hypsodonty. *Palaeogeogr. Palaeoclimatol. Palaeoecol.* **177**, 59-75 (2002).
- Quan, C., Liu, Y.C. & Utescher, T. Eocene monsoon prevalence over China: A paleobotanical perspective. *Palaeogeogr. Palaeoclimatol. Palaeoecol.* **365-366**, 302-311 (2012).
- Spicer, R.A. *et al.* Asian Eocene monsoons as revealed by leaf architectural signatures. *Earth Planet. Sc. Lett.* **449**, 61-68 (2016).
- Caves Rugenstein, J.K. & Chamberlain, C.P. The evolution of hydroclimate in Asia over the Cenozoic: A stable-isotope perspective. *Earth-Sci. Rev.* **185**, 1129-1156 (2018).
- Guo, Z.T. *et al.* Onset of Asian desertification by 22 Myr ago inferred from loess deposits in China. *Nature* **416**, 159-163 (2002).
- Nie, J. *et al.* Pacific freshening drives Pliocene cooling and Asian monsoon intensification. *Sci. Rep.* **4**(2015).
- Bougeois, L. *et al.* Asian monsoons and aridification response to Paleogene sea retreat and Neogene westerly shielding indicated by seasonality in Paratethys oysters. *Earth Planet. Sci. Lett.* **485**, 99-110 (2018).

- Page, M. *et al.* Synchronous cooling and decline in monsoonal rainfall in northeastern Tibet during the fall into the Oligocene icehouse. *Geology* **47**, 203-206 (2019).
- An, Z. *et al.* Global Monsoon Dynamics and Climate Change. *Annu. Rev. Earth Planet. Sci.* **43**, 29-77 (2015).
- Wang, P.X. *et al.* The global monsoon across time scales: Mechanisms and outstanding issues. *Earth-Sci. Rev.* **174**, 84-121 (2017).
- Jiang, H. & Ding, Z. A 20 Ma pollen record of East-Asian summer monsoon evolution from Guyuan, Ningxia, China. *Palaeogeogr. Palaeoclimatol. Palaeoecol.* **265**, 30-38 (2008).
- Lu, H., Wang, X. & Li, L. Aeolian dust records indicate the linkage of global cooling and Asian drying in late Cenozoic. *Quaternary Sciences* **28**, 949-956 (2008).
- Lu, H., Wang, X. & Li, L. Aeolian sediment evidence that global cooling has driven late Cenozoic stepwise aridification in central Asia. *Geological Society, London, Special Publications* **342**, 29-44 (2010).
- Lu, H. & Guo, Z. Evolution of the monsoon and dry climate in East Asia during late Cenozoic: A review. *Science China Earth Sciences* **57**, 70-79 (2014).
- Lu, H. *et al.* Cenozoic depositional sequence in the Weihe Basin (Central China): A long-term record of Asian monsoon precipitation from the greenhouse to icehouse Earth. *Quaternary Sciences* **38**, 1057-1067 (2018).
- Ripley, B. *et al.* Fire ecology of C₃ and C₄ grasses depends on evolutionary history and frequency of burning but not photosynthetic type. *Ecology* **96**, 2679-2691 (2015).
- Taylor, S.H. *et al.* CO₂ availability influences hydraulic function of C₃ and C₄ grass leaves. *J. Exp. Bot.* **69**, 2731-2741 (2018).

- Wang, W. *et al.* A revised chronology for Tertiary sedimentation in the Sikouzi basin: Implications for the tectonic evolution of the northeastern corner of the Tibetan Plateau. *Tectonophysics* **505**, 100-114 (2011).
- Lin, X., Chen, H., Wyrwoll, K. & Cheng, X. Commencing uplift of the Liupan Shan since 9.5Ma: Evidences from the Sikouzi section at its east side. *J. Asian Earth Sci.* **37**, 350-360 (2010).
- Zheng, D. *et al.* Rapid exhumation at ~8 Ma on the Liupan Shan thrust fault from apatite fission-track thermochronology: Implications for growth of the northeastern Tibetan Plateau margin. *Earth Planet. Sci. Lett.* **248**, 198-208 (2006).
- Cotton, J.M., Sheldon, N.D. & Strömberg, C.A.E. High-resolution isotopic record of C₄ photosynthesis in a Miocene grassland. *Palaeogeogr. Palaeoclimatol. Palaeoecol.* **337-338**, 88-98 (2012).
- Chen, S.T., Smith, S.Y., Sheldon, N.D. & Strömberg, C.A.E. Regional-scale variability in the spread of grasslands in the late Miocene. *Palaeogeogr. Palaeoclimatol. Palaeoecol.* **437**, 42-52 (2015).
- Harris, E.B., Strömberg, C.A.E., Sheldon, N.D., Smith, S.Y. & Vilhena, D.A. Vegetation response during the lead-up to the middle Miocene warming event in the Northern Rocky Mountains, USA. *Palaeogeogr. Palaeoclimatol. Palaeoecol.* **485**, 401-415 (2017).

----- (The end)

Reviewers' comments:

Reviewer #1 (Remarks to the Author):

As stated in my previous review, this paper provides the first phytolith-based record of vegetation change in eastern Eurasia (specifically China) during the late Miocene-Pleistocene. The data presented by the Wang et al. convincingly demonstrates a shift from C4 (or PACMAD) dominated grasslands to pooid-dominated grasslands at ~4 Ma, pointing to cooling and drying during this time. These results should be of interest to a wide audience of biologists and geologists.

The paper is well written and researched and very appropriate for Nature Communications. The authors addressed all my previous concerns. After the revisions, I have only two main comments (and some minor ones, see below):

- The authors use min and max estimations of C4 grasses (respectively, panicoids + chloridooids vs. all PACMAD grasses), but when discussing grass communities, seem to interpret all of the PACMAD types as C4. As far as I can tell, they present no independent evidence to support that all of their PACMAD morphotypes are from C4 grasses. This should be more clearly taken into account when discussing their results and the implications thereof.
- The English could use more work, especially in the Supplementary Materials. I did not go through and edit, but my minor comments below give examples where the text is hard to follow as a result. I encourage the authors to go through and carefully correct the English.

Minor comments; ms:

Line 55: instead of "Panicoideae-Aristidoideae-Chloridoideae-..." write "Panicoideae, Aristidoideae, Chloridoideae, Centothecoideae, Micrairoideae, Arundinoideae, and Danthonioideae"

Line 58: cite Edwards and Smith (2010) for the results described in this sentence (ending with "tend to be cold-adapted.")

Line 177: There is no strong evidence that grasses at Shanwang were C4. Therefore, insert the word "potential" so that it reads "... early Miocene sediments in Shanwang in northern China indicate that potential C4 grasses were...".

Line 194: Replace the word "by" with "using". Also, the pattern of C3 grassy habitats preceding C4 grasslands was discussed at length by Edwards et al. (2010) and Strömberg (2011). These references should be mentioned.

Line 333: There was only one change in the classification (as I pointed out in the first review), namely the addition of the scooped bilobate/cross of Oryzoideae. The collapsed saddle was included in the classification of Strömberg and McInerney (2011: Table 3).

Fig. 3a: "FI-ratio" should be "FI-t ratio". "Close-habitat grass" should be "Closed-habitat grass".

Supp Mat:

Line 120-121: It should be "BO clade", not "BOclade".

Line 144-onward: This sentence makes no sense, please rewrite: "However, the climate range of each subfamily calculated from the union of climate range of all its species is obvious approach its tolerance limit or beyond its ecological amplitude."

Line 155-onward: change the sentence fragment: "the optimum range of each species in which species can growth and reproduction is the best choice however difficult to obtain" to "the optimum range of each species in which species can grow and reproduce is the best choice however difficult to obtain"

Line 147: "obvious" should be "obviously".

Table S1: There is just one modification. See my comment above.

Caroline Strömberg

Reviewer #2 (Remarks to the Author):

Dear authors,

I went through the revised version of your ms. I was happy to see that you carefully addressed all points raised by the reviewers. From my point of view no more changes are required, and I'm looking forward to see the ms published.

With kind regards,
Torsten Utescher

Reviewer #3 (Remarks to the Author):

There are a number of issues with the manuscript that need to be addressed by the authors.

Line 24: landform growth forcing hypothesis. What does this hypothesis predict? It is not mentioned later in the text, or I missed it.

Lines 34, 35: Ref. 1 - Didn't these authors say increase of C4 plants was coupled with open vegetation which in turn was coupled with increasing aridity.
If you look at climatic patterns in this area, you will see that small scale mosaics of summer wet and summer dry climates exist.

Line 38: leaf wax - yes, of course, here it might have been an early form of Mediterranean climate.

Lines 44-46: See my comment below: How reliable is the monsoon proxy "C4 expansion"?

Lines 63-64: Yes, but this is only possible if these niches did not shift since 11 million years (in case of your study).

Line 66: "During this expansion" - unclear what is meant here, geographical or temporal expansion? Please clarify.

This whole sentence is a bit awkward. Please re-phrase.

Line 74: " Here, we explore the history of C4 grasses and the monsoon climate in East Asia..."
This is my main concern with this paper: Are these the same histories???

There is no evidence for this being the case provided in this study.

Line 89: How much bias do you expect for the phytolith assemblage?
Compare for example the paper by Bremond et al. 2004 Review Paleobotany Palynology

Lines 96-98: This appears to be rather trivial, the cooling would be enough to explain decreasing C4 grasses.

Lines 124-126: Ok, how likely is this?

Line 167, and following section: Fine, but what does this tell about monsoon?

Earliest record of C4 grasses in East Asia - but this does not appear to be coupled with the onset of monsoon in this area. See references in this section and see records of chemical weathering from the South China Sea indicating a strengthening of E Asian monsoon after c. 24 Ma and a peak during the middle Miocene. Same is true for hematite/goethite ratios. These and other data also indicate weakening of monsoon during the Pliocene (see. e.g. Clift 2006 Earth Planet Sci Lett; Clift et al., 2008 Nature Geosciences; Clift et al., 2014 Earth-Science Rev).

Therefore, I wonder what the significance of a decrease in C4 grasses in the Pliocene is, when

Earth cooled significantly.

Lines 212ff: This is very interesting.

Lines 244-255: Didn't you say before it was all grass-dominated???

Lines 376, 377: Unclear, please re-phrase.

Lines 377, 378: "... these data are not available for open access". What does this mean? Please explain.

Reviewer #2 (Additional Remarks to the Author):

I went again through the points of Reviewer 3. It also seems to me that the most critical point is the interpretation of past monsoonal conditions. As regards points 2, 7, and 15 of Reviewer 3 (version with replies) I agree with Reviewer3: Monsoon is a seasonal phenomenon and therefore it is impossible to estimate monsoon intensity using annual means. Hence, based on annual values, only assumptions can be made while there is no clear evidence.

I suppose that in principle this is clear to the authors because e.g. in lines 266-268 they state: "The decrease in temperature was probably influenced by global cooling, and the decrease in precipitation was probably influenced by a decrease in Asian monsoon rainfall."

To avoid irritations it would be good to be as careful with MAP interpretations in terms of monsoon, also elsewhere in the ms, preferably add the statement that MAP cannot be regarded as a clear indicator for monsoon intensity.

To estimate MAP, the authors apply the Coexistence Approach on the recovered phytolith taxa (line 366 and following). As a standard, also monthly precipitation data are reconstructed by the CA (warm, dry, wet month precipitation). I suggest to reconstruct these variables for the presently studied material. As I understand it these data can be easily obtained from the climatic envelopes of the Poaceae taxa already compiled by the authors. The consideration of monthly means would definitely improve evidence for the existence of a monsoonal climate and could also lead a better quantification of monsoon intensity. A calculation of the seasonality of precipitation using the CV (coefficient of variation) of the precipitation of twelve months following Edward and Smith, 2010, PNAS, as done by the authors, is not really useful in the present context as I think.

Maybe it would be good to ask the authors to add a couple of considerations in the actual round as regards the causal relationship of monsoonal climate history and evolution of C4 grasslands. As in many other cases a clear proof of this relationship should not be possible. Nevertheless, there is some evidence that both factors are related. A couple of minor points mentioned by Reviewer 3 can be easily addressed by the authors.

I think the paper should be accepted on a satisfying reply on the points raised in the present round. I still think the ms is a valuable contribution stimulating the ongoing discussion of the topic.

Response to the key question:

The annual rainfall can be used as an indicator of the East Asian summer monsoon (EASM) for the following reasons:

1. The EASM is closely associated with annual rainfall by its definition. In meteorology, an important definition of monsoon conditions is closely related to its rainfall: “*Annual range exceeds 300 mm (or 2 mm/day), local summer precipitation exceeds 55% of the annual total precipitation*” (Wang, 1994; Wang and Ding, 2008). There are too many criteria used to define the intensity of monsoons (Wang et al., 2008; Zhao et al., 2015). In terms of the EASM, the increased wind strength is not always coupled with the increase in monsoon precipitation on the decadal timescale, and the precipitation shows spatial diversity (“southern China flood and northern China drought”) (Ding et al., 2008). The reason for this phenomenon is complicated and could be related to Pacific Decadal Oscillation (PDO), Atlantic multidecadal oscillation (AMO) and Indian Ocean Basin mode (IOBM) (e.g., Zhang et al., 2018, *J. Climate*). Taken together, the definition of monsoons is still under debate among meteorologists. For geologists and palaeoclimatologists, we tend to use the mean annual precipitation (MAP) as the definition of monsoons. However, we should be careful with the words such as “monsoon intensity” and “strengthening/weakening of the monsoon”, because of the possibility of uncoupled wind strength and precipitation, instead, we use “increased/decreased monsoon precipitation/rainfall” to describe the monsoon intensity evolution in our manuscript. Moreover, we use both of the summer time precipitation and the MAP as an indicator of the monsoon strength, since the two were closely associated with each other at the long timescale (Bin Wang, 2006. *The Asian monsoon*. Springer, 2006).

In our study area, local summer precipitation (June to September) accounts for over 60% of the annual total precipitation (please see Figure 1 below); hence, it is a typical monsoon climate by definition. The monsoon circulation has probably remained

similar to the current conditions since the Eocene (e.g., Quan et al., 2012, Palaeo3; Spicer et al., 2016, EPSL; Lu et al., 2018, Quaternary Sciences; Caves and Chamberlain, 2018, ESR), or Miocene (An et al., 2001, Nature; Guo et al., 2002, Nature; Wang et al., 2017, ESR; Clift, 2006 EPSL; Clift et al., 2008 Nat. Geosci.; Clift et al., 2014 ESR). These studies suggested that our study area has been a monsoon-dominated climate since at least ~11 Ma. Therefore, using MAP to indicate monsoon rainfall is reasonable. From another point of view, the moisture source of the precipitation in our study area mainly (80%) originated from the monsoon dominated region (see Figure 2 below). Few of the moisture sources from the northwest (20%) originated from the westerlies, which means that the precipitation was mainly caused by the summer monsoon. Therefore, the mean annual precipitation could be used as an indicator of the summer monsoon strength.

Figure 1. Modern climate of Lantian, data source <http://data.cma.cn/>, calculated from the period of 198101-201012.

Figure 2. Water source analyses of precipitation at our study site, Lantian, Xian. Grouped by cluster analysis into six distinct trajectory directions: 1, 2 and 6 from the northwest, 3 and 5 from the summer monsoon and 4 from local. Amount (mm) of monthly precipitation that falls in the Lantian region as a result of the 6 trajectories.

2. From a geological perspective, the definition of monsoon intensity is different

from the meteorological perspectives. At millennial to orbital time scales, a strengthened EASM was indicated by the increased precipitation (both year round or seasonal). For example, studies of alternating layers of eolian dust and soils in the Chinese Loess Plateau suggest periods of stronger and weaker EASM (e.g., An et al., 2015, *Annual Review of Earth and Planetary Sciences*). Studies of marine sediments in the South China Sea reveal millennial-scale summer monsoon variability (e.g., Oppo and Sun, 2005, *Geology*; Wang et al., 2017, *ESR*). These studies used the amount of MAP to indicate the intensity of EASM.

At the tectonic time scale, the amount of precipitation year round is regarded as a direct indicator of the monsoon strength (Licht et al., 2014, *Nature*; Roe et al., 2016, *JGR*; Huber and Goldern, 2012; Spicer et al., 2016). For example, the studies of ODP 1148 in the South China Sea reveal variations in the EASM since ~25 Ma (Clift 2006 *EPSL*; Clift et al., 2008 *Nat. Geosci.*; Clift et al., 2014, *ESR*). These studies used the MAP to indicate the monsoon strength, too.

3. Palaeoclimatic modelling results show that the strengthened monsoon circulation was associated with increasing monsoon rainfall at the tectonic timescale, such as Eocene (Licht et al., 2014, *Nature*; Li et al., 2017, 2018, *Palaeo3*), Miocene (Herold et al., 2011, *J. Clim*) and Pliocene (Yan et al., 2012, *Atmospheric and Oceanic Science Letters*; Zhang et al., 2013, *Clim. Past*). The simulation of the mid-Pliocene East Asian monsoon climate shows that 13 of the 15 models show increased East Asian summer wind as well as increased mean annual precipitation and summer precipitation compared to the preindustrial level (Zhang et al., 2013, *Clim. Past*), which strongly supports that the mean annual precipitation can be used as an indicator of EASM. The coupled strength of EASM and MAP can also be found at the orbital time scale (e.g., Kutzbach et al., 2008, *Climate Dynamics*) and the millennial time scale (Lu et al., 2013, *Geology*; Liu et al., 2014, *QSR*). Notice that this coupled relationship is different at the decadal timescale suggested by meteorological data (see discussion above). One of the most important roles of the palaeoclimatic model is

to test the reliability of geological records. There are few models that discuss the seasonality of precipitation, except for some (Li et al., 2018, Palaeo3), because there are few geological proxies that can be used to describe the seasonality of precipitation.

4. From a biological perspective, when using fossil records, such as pollen, phytoliths and microfossils, to quantitatively reconstruct paleoclimate, the modern reference used must be within a certain range of the study site to obtain reliable results (Cao et al., 2017, QSR). For the EASM region, the reconstructed monsoon climate is concluded as long as the sites are located in the EASM region. In addition, we use the precipitation of the warmest month (WMMP) and the warm season (April to September) precipitation (WSP), which are seasonal rainfalls in summer time, to indicate the monsoon rainfall. These newly added results support our conclusions that the monsoon precipitation decreased during the Pliocene (Table S5). Therefore, we suggest that the MAP is directly related to the strength of monsoon circulation.

Table S5. Reconstructed climate parameters and the comparison with the modern climate of Lantian, calculated by the coexistence approach and ecosystem matching.

	Weather station (Weinan)	CA			CA and ecosystem matching		
		11.0~4.2 Ma	4.2~2.6 Ma	2.6~0 Ma	11.0~4.2 Ma	4.2~2.6 Ma	2.6~0 Ma
MAT (°C)	13.8	9.7~15.3	9.7~15.3	3.3~15.3	11~15.3	9.7~11	3.3~11
WMMT (°C)	26.8	18.9~26.4	18.9~26.4	13.8~26.4	20~26.4	18.9~20	13.8~20
CMMT (°C)	-0.3	0.5~9	0.5~9	-8.9~9	2~9	0.5~2	-8.9~2
MAP (mm)	569	812~1673	812~1673	441~1673	812~1673	812~900	441~900
WSP(mm)	439	417~1197	417~1197	263~1197	427~1197	417~540	263~540
WMMP (mm)	95	76~199	51~199	43~257	76~199	76~130	43~130
CMMP (mm)	6	36~120	36~120	6~120	36~120	36~74	6~74
DT (°C)	27.1	10~22.8	10~24.1	10~24.1	10~18	15~24.1	15~24.1

For the above reasons, we suggest that the MAP can be used as an indicator of the EASM. We added explanations in the manuscript (line 157-159; line 262-267; line 275-279).

Reviewer #1

1) The authors use min and max estimations of C4 grasses (respectively, panicoids + chloridoids vs. all PACMAD grasses), but when discussing grass communities, seem to interpret all of the PACMAD types as C4. As far as I can tell, they present no independent evidence to support that all of their PACMAD morphotypes are from C4 grasses. This should be more clearly taken into account when discussing their results and the implications thereof.

Response: Thank you for this suggestion. We are also aware that not all of the PACMAD types are C4; therefore, we added some interpretations (minimum and maximum estimates) to the manuscript when discussing C4 grasses, such as:

Line 172: Our results reveal that C4 grasses were moderately (minimum estimate) to very abundant (maximum estimate) in the study area at ~11.0 Ma.

Line 195: Our data suggest that a marked decrease in C4 grasses (both maximum and minimum estimates) occurred at ~4.2 Ma.

Line 205: A controversial issue is that our phytolith data provide a different estimate of C₄ grasses (both maximum and minimum estimates) than the estimate derived from $\delta^{13}\text{C}$ data of soil carbonates.

For samples younger than ~130 ka, we did not discuss the maximum and minimum estimates of C4 grasses because there are no components of PACMAD in general that result in the same maximum and minimum estimates of C4 grasses.

2) The English could use more work, especially in the Supplementary Materials. I did not go through and edit, but my minor comments below give examples where the text is hard to follow as a result. I encourage the authors to go through and carefully correct the English.

Response: We apologize for the linguistic problem. We have performed a careful correction of the English, especially in the Supplementary Information. In addition, we have invited native English speakers to polish the English writing. We hope you are satisfied with the revised manuscript and Supplementary Information.

3) Minor comments; ms: Line 55: instead of “Panicoideae-Aristidoideae-Chloridoideae-...” write “Panicoideae, Aristidoideae, Chloridoideae, Centothecoideae, Micrairoideae, Arundinoideae, and Danthonioideae”

Response: This change has been made.

4) Line 58: cite Edwards and Smith (2010) for the results described in this sentence (ending with “tend to be cold-adapted.”)

Response: This change has been made.

5) Line 177: There is no strong evidence that grasses at Shanwang were C4. Therefore, insert the word “potential” so that it reads “... early Miocene sediments in Shanwang in northern China indicate that potential C4 grasses were...”.

Response: This change has been made.

6) Line 194: Replace the word “by” with “using”. Also, the pattern of C3 grassy habitats preceding C4 grasslands was discussed at length by Edwards et al. (2010) and Strömberg (2011). These references should be mentioned.

Response: This change has been made.

7) Line 333: There was only one change in the classification (as I pointed out in the first review), namely the addition of the scooped bilobate/cross of Oryzoideae. The collapsed saddle was included in the classification of Strömberg and McInerney (2011: Table 3).

Response: We are sorry for the mistake; this change has been made.

8) Fig. 3a: “FI-ratio” should be “FI-t ratio”. “Close-habitat grass” should be “Closed-habitat grass”.

Response: Corrected.

9) Supp Mat: Line 120-121: It should be “BO clade”, not “BOclade”.

Response: Corrected.

10) Line 144-onward: This sentence makes no sense, please rewrite: “However, the climate range of each subfamily calculated from the union of climate range of all its species is obvious approach its tolerance limit or beyond its ecological amplitude.”

Response: Thank you. We have rewritten this sentence: “However, the climate range of each subfamily calculated from the combination of climate ranges of all its species was too wide, and this calculated climate range could approach a subfamily’s tolerance limit or exceed its ecological amplitude.”

11) Line 155-onward: change the sentence fragment: “the optimum range of each species in which species can growth and reproduction is the best choice however difficult to obtain” to “the optimum range of each species in which species can grow and reproduce is the best choice however difficult to obtain”

Response: This change has been made.

12) Line 147: “obvious” should be “obviously”.

Response: Corrected.

13) Table S1: There is just one modification. See my comment above.

Response: This change has been made.

Reviewer #3

1) Line 24: landform growth forcing hypothesis. What does this hypothesis predict? It is not mentioned later in the text, or I missed it.

Response: Thank you for the question. The landform growth forcing hypothesis predicts that with the growth of the Tibetan Plateau, the Asian monsoon circulation became stronger, which brought more precipitation inland (Kutzbach et al., 1989; Prell and Kutzbach, 1992; An et al., 2001). We mentioned it in line 279: “Several possible drivers have been proposed to explain Asian climate change since the late Miocene, such as the growth of the Tibetan Plateau and global cooling.”

2) Lines 34, 35: Ref. 1 - Didn't these authors say increase of C4 plants was coupled with open vegetation which in turn was coupled with increasing aridity.

If you look at climatic patterns in this area, you will see that small scale mosaics of summer wet and summer dry climates exist.

Response: No, these authors did not mean increasing aridity. Ref. 1 and 2 suggest that the expansion of C4 grasses indicates the development of a seasonal climate and increased warm-season precipitation rather than an increase in aridity at ~8 Ma. These authors' opinions are quite different from those of Ref. 3, which suggests that an overall increase in aridity was the trigger for this vegetation change.

For your comment, yes, there is a small-scale mosaic pattern of wet and dry existing at meteorological timescale in this area, but this is not the case at glacial-interglacial timescales and the longer. Because the climate has varied on multiple timescales over the late Cenozoic, we cannot use the short timescale climate pattern to interpret the long timescale climatic variations. This conclusion has been confirmed by many studies.

Please see the beginning of the response section.

3) Line 38: leaf wax - yes, of course, here it might have been an early form of Mediterranean climate.

Response: We thought the subtext of Ref. 3 is that the monsoon climate in South Asia was strong enough to bring abundant precipitation for the growth of trees before ~8 Ma. The monsoon climate became weaker after ~8 Ma, and reduced summer precipitation and/or increased winter precipitation resulted in open vegetation and the expansion of C4 grasses.

4) Lines 44-46: See my comment below: How reliable is the monsoon proxy “C4 expansion”?

Response: Thank you for the question. “...the use of stable carbon isotopic composition as a proxy for the expansion of C4 grasses and therefore monsoon precipitation” is also what we questioned because “the drivers of C4 expansion and the environmental niches of C4 grasses are still unclear”. The next paragraph (line 50-73) examines this question. We found that the key to solving this problem is to determine which type of C4 grasses expanded, and our phytolith assemblages are appropriate to solve this problem.

5) Lines 63-64: Yes, but this is only possible if these niches did not shift since 11 million years (in case of your study).

Response: Thank you for the comment. You raised the same question in your last review (comment 27). The assumption we made is that the environmental niches of NLRs are the same as fossil species on a regional scale, which means that there is no mass migration of species across the continent (e.g., from North America to Asia) while ignoring the possible changes in environmental niches affected by the evolution process. We added information on the assumptions and potential sources of error to the manuscript (please see page 6, lines 124~126): “However, this quantitative reconstruction is valid only if we make the assumption that the same species thrived

since the late Miocene and that the environmental niches of these grass subfamilies have not evolved.”

6) Line 66: ”During this expansion” – unclear what is meant here, geographical or temporal expansion? Please clarify.

This whole sentence is a bit awkward. Please re-phrase.

Response: Thank you. We mean temporal expansion, and we rephrased this sentence: “If it were possible to evaluate the ecological role (from non-dominant to dominant) played by each clade of grasses during the expansion of C₄ grasses, then the corresponding environmental controls and driving mechanisms could be identified...”

7) Line 74: ” Here, we explore the history of C4 grasses and the monsoon climate in East Asia...”

This is my main concern with this paper: Are these the same histories????

There is no evidence for this being the case provided in this study.

Response: Thank you for this question. We are sorry for this mistake, “explore the history of C₄ grasses” is indeed inappropriate; we changed “C₄ grasses” to “vegetation”.

8) Line 89: How much bias do you expect for the phytolith assemblage?

Compare for example the paper by Bremond et al. 2004 Review Paleobotany Palynology

Response: Thank you for the question. The bias of phytolith assemblages could be obtained through comparison with other proxies, such as pollen data. We discussed this in detail in the manuscript, line 206~216, “To test the validity of our estimate...”

9) Lines 96-98: This appears to be rather trivial, the cooling would be enough to explain decreasing C4 grasses.

Response: Thank you for the comment. The ultimate goal of this paper is to reconstruct past monsoon rainfall variation, not to explain why C4 grasses decreased. The decrease in C4 grasses is just a part of the vegetation change obtained from our phytolith assemblages. We also obtain vegetation structure and grass community from phytolith assemblages. Assessing these vegetation changes together, we quantitatively reconstructed past monsoon rainfall variation, which is very important to understanding the long-term monsoon variations and associated forcing mechanisms.

10) Line 167, and following section: Fine, but what does this tell about monsoon?

Earliest record of C4 grasses in East Asia – but this does not appear to be coupled with the onset of monsoon in this area. See references in this section and see records of chemical weathering from the South China Sea indicating a strengthening of E Asian monsoon after c. 24 Ma and a peak during the middle Miocene. Same is true for hematite/goethite ratios. These and other data also indicate weakening of monsoon during the Pliocene (see. e.g. Clift 2006 Earth Planet Sci Lett; Clift et al., 2008 Nat. Geosci.; Clift et al., 2014 Earth-Science Rev).

Therefore, I wonder what the significance of a decrease in C4 grasses in the Pliocene is, when Earth cooled significantly.

Response: Thank you for the question and comment. For your question, line 167 and the following section discuss the decrease in C4 grasses and explain the inconsistency with other proxies. We did not discuss the implication of C4 grasses for monsoons in this part because C4 grasses are only a part of the vegetation change obtained from our phytolith assemblages. We discuss monsoon precipitation in the next section (line 240 and following), which is directly derived from our phytolith assemblages.

For on your comment, we said our phytolith record is “most likely the earliest fossil record of C4 grassland in East Asia” but did not talk about its implication for monsoon climate. The onset of monsoon is another story and is still under debate (see our detailed response to that **MAP could be used as an indicator for EASM**); the record of C4 grasses did not have to be coupled with the monsoon evolution. Moreover, we are confused about the reason you mentioned Peter Clift’s work in South China Sea: if you mean these studies suggest the early onset of Asian monsoon, we would like to say the record of C4 grasses did not have to be coupled with monsoon evolution; if you mean these work has already reconstructed Asian monsoon variation and was the same as our result, while we would like to say that we do not using C4 grasses to directly reconstruct Asian monsoon (see our response to your last comment).

11) Lines 212ff: This is very interesting.

Response: Thank you.

12) Lines 244-255: Didn’t you say before it was all grass-dominated???

Response: Thank you for the question. We suggest it is quite normal that there are some mosaic of trees and woodland in the grassland. Although we mentioned the “resumption of woodland” suggested by Ref. 16 and the “presence of species that lived in a closed canopy” suggested by Ref. 17, the percentage of woodland in the flora and the percentage of species that lived in closed canopy were very low, which is consistent with our results.

13) Lines 376, 377: Unclear, please re-phrase.

Response: Thank you. The modern analogue matching procedure refers to the “modern analogue method”, which is commonly used in pollen-based quantitative reconstruction of paleoclimate (Birks et al., 2010, The Open Ecology Journal 3, 68-110). We add this to the manuscript.

14) Lines 377, 378: "... these data are not available for open access". What does this mean? Please explain.

Response: These data exist but are not possible to share due to national security restrictions because these data contain sensitive information, such as the distribution of national protected plants, endangered plants and other plants (poppy).

Response to the other reviewer

The other reviewer has three main comments and suggestions:

1) To avoid irritations it would be good to be as careful with MAP interpretations in terms of monsoon, also elsewhere in the ms, preferably add the statement that MAP cannot be regarded as a clear indicator for monsoon intensity.

Response: Thank you for the suggestion. To address this issue, we provide a detailed response in the beginning of the response section as "**MAP could be used as an indicator for EASM**". We also discussed it briefly in the manuscript following your suggestion (line 157-159; line 262-267; line 275-279). In the new version, we added data of the warmest month precipitation (WMMP) and the warm season (April to September) precipitation (WSP), which are indicators of monsoon precipitation, to reconstruct the Asian monsoon strength. These results also support our conclusions.

2) To estimate MAP, the authors apply the Coexistence Approach on the recovered phytolith taxa (line 366 and following). As a standard, also monthly precipitation data are reconstructed by the CA (warm, dry, wet month precipitation). I suggest to reconstruct these variables for the presently studied material. As I understand it these data can be easily obtained from the climatic envelopes of the Poaceae taxa already compiled by the authors. The consideration of monthly means would definitely improve evidence for the existence of a monsoonal climate and could also lead a better quantification of monsoon intensity. A calculation of the seasonality of precipitation using the CV (coefficient of variation) of the precipitation of twelve

months following Edward and Smith, 2010, PNAS, as done by the authors, is not really useful in the present context as I think.

Response: Thank you very much; this is a very good idea. We have reconstructed the WMMP and WSP as indicators of the monsoon strength, and these two parameters show decreasing monsoon strength at ~4.2 Ma and ~2.6 Ma, respectively. In addition, the WSP dominated the MAP, suggesting that a monsoon climate existed since ~11 Ma.

3) Maybe it would be good to ask the authors to add a couple of considerations in the actual round as regards the causal relationship of monsoonal climate history and evolution of C4 grasslands. As in many other cases a clear proof of this relationship should not be possible. Nevertheless, there is some evidence that both factors are related.

Response: Thank you very much. This is a very good idea. Many papers have studied the relationship between monsoon precipitation and C4 grasses in the Chinese Loess Plateau (An et al., 2005, Geology; Liu et al., 2005; Yang et al., 2015, PNAS). The conclusion is that strengthened monsoon precipitation can enhance the growth of C4 plants in this region; therefore, the C4 grasses are coupled with the monsoon intensity in Pleistocene glacial-interglacial cycles in our study region. However, the relationship between monsoon precipitation and C4 grasses is complicated in the warmer world during the Miocene and Pliocene, in which it seems that more arboreal plants were involved in this semiarid and semi-humid region. Thus, a new interpretation is necessary to explain the positive shift of $\delta^{13}\text{C}_{\text{SC}}$ data and climate change, and we offered a new interpretation in this study. Please see the discussion section of the manuscript (line 217~238).

References cited

- An, Z. *et al.* Global Monsoon Dynamics and Climate Change. *Annu. Rev. Earth Planet. Sci.* **43**, 29-77 (2015).
- An, Z.S., Kutzbach, J.E., Prell, W.L. & Porter, S.C. Evolution of Asian monsoons and phased uplift of the Himalayan Tibetan plateau since Late Miocene times. *Nature* **411**, 62-66 (2001).
- An, Z. *et al.* Multiple expansions of C₄ plant biomass in East Asia since 7 Ma coupled with strengthened monsoon circulation. *Geology* **33**, 705-708 (2005).
- Birks, H. J. B., Heiri, O., Seppä, H and Bjune, A, E., 2011. Strengths and weaknesses of quantitative climate reconstructions based on late-Quaternary biological proxies. *The Open Ecology Journal* **3**, 68-110 (2011).
- Cao, X. *et al.* Impacts of the spatial extent of pollen-climate calibration-set on the absolute values, range and trends of reconstructed Holocene precipitation. *Quaternary Sci. Rev.* **178**, 37-53 (2017).
- Caves Rugenstein, J.K. & Chamberlain, C.P. The evolution of hydroclimate in Asia over the Cenozoic: A stable-isotope perspective. *Earth-Sci. Rev.* **185**, 1129-1156 (2018).
- Clift, P.D. Controls on the erosion of Cenozoic Asia and the flux of clastic sediment to the ocean. *Earth Planet. Sc. Lett.* **241**, 571-580 (2006).
- Clift, P.D. *et al.* Correlation of Himalayan exhumation rates and Asian monsoon intensity. *Nat. Geosci.* **1**, 875-880 (2008).
- Clift, P.D., Wan, S. & Blusztajn, J. Reconstructing chemical weathering, physical erosion and monsoon intensity since 25Ma in the northern South China Sea: A review of competing proxies. *Earth-Sci. Rev.* **130**, 86-102 (2014).
- Ding, Y., Wang, Z. & Sun, Y. Inter-decadal variation of the summer precipitation in

- East China and its association with decreasing Asian summer monsoon. Part I: Observed evidences. *Int. J. Climatol.* **28**, 1139-1161 (2008).
- Guo, Z.T. *et al.* Onset of Asian desertification by 22 Myr ago inferred from loess deposits in China. *Nature* **416**, 159-163 (2002).
- Herold, N., Huber, M., Müller, R.D., 2011. Modeling the Miocene climatic optimum. Part I: Land and atmosphere. *J. Clim.* **24**, 6353–6372.
- Kutzbach J E, Guetter P J, Ruddiman W F, et al. Sensitivity of climate to Late Cenozoic uplift in Southern Asia and the American West: Numerical experiments. *J Geophys. Res.* **94**, 18393–18407 (1989).
- Kutzbach, J.E., Liu, X., Liu, Z. & Chen, G. Simulation of the evolutionary response of global summer monsoons to orbital forcing over the past 280,000 years. *Clim. Dynam.* **30**, 567-579 (2008).
- Li, X., Zhang, R., Zhang, Z., Yan Q., 2017. What enhanced the aridity in Eocene Asian inland: Global cooling or early Tibetan Plateau uplift? *Palaeogeogr. Palaeoclimatol. Palaeoecol.* <http://dx.doi.org/10.1016/j.palaeo.2017.10.029>.
- Li, X., Zhang, R., Zhang, Z., Yan Q., 2018. Do climate simulations support the existence of East Asian monsoon climate in the Late Eocene? *Palaeogeogr. Palaeoclimatol. Palaeoecol.* <https://doi.org/10.1016/j.palaeo.2017.12.037>.
- Licht, A. *et al.* Asian monsoons in a late Eocene greenhouse world. *Nature* **513**, 501-506 (2014).
- Liu, W. *et al.* Summer monsoon intensity controls C4/C3 plant abundance during the last 35 ka in the Chinese Loess Plateau: Carbon isotope evidence from bulk organic matter and individual leaf waxes. *Palaeogeography, Palaeoclimatology, Palaeoecology* **220**, 243-254 (2005).

- Liu, Z. *et al.* Chinese cave records and the East Asia Summer Monsoon. *Quaternary Sci. Rev.* **83**, 115-128 (2014).
- Lu, H. & Guo, Z. Evolution of the monsoon and dry climate in East Asia during late Cenozoic: A review. *Science China Earth Sciences* **57**, 70-79 (2014).
- Lu, H. *et al.* Cenozoic depositional sequence in the Weihe Basin (Central China): A long-term record of Asian monsoon precipitation from the greenhouse to icehouse Earth. *Quaternary Sciences* **38**, 1057-1067 (2018).
- Lu, H., Wang, X. & Li, L. Aeolian dust records indicate the linkage of global cooling and Asian drying in late Cenozoic. *Quaternary Sciences* **28**, 949-956 (2008).
- Lu, H., Wang, X. & Li, L. Aeolian sediment evidence that global cooling has driven late Cenozoic stepwise aridification in central Asia. *Geological Society, London, Special Publications* **342**, 29-44 (2010).
- Oppo, D.W. & Sun, Y. Amplitude and timing of sea-surface temperature change in the northern South China Sea: Dynamic link to the East Asian monsoon. *Geology* **33**, 785-788 (2005).
- Prell, W.L., Kutzbach, J.E. Sensitivity of the Indian monsoon to forcing parameters and implications for its evolution. *Nature* **360**, 647-652 (1992).
- Quan, C., Liu, Y.C. & Utescher, T. Eocene monsoon prevalence over China: A paleobotanical perspective. *Palaeogeogr. Palaeoclimatol. Palaeoecol.* **365-366**, 302-311 (2012).
- Ramage, C.S. *Monsoon Meteorology*. Academic Press, New York (1971).
- Spicer, R.A. *et al.* Asian Eocene monsoons as revealed by leaf architectural signatures. *Earth Planet. Sc. Lett.* **449**, 61-68 (2016).
- Wang, B. & Ding, Q. Global monsoon: Dominant mode of annual variation in the

- tropics. *Dynam. Atmos. Oceans* 44, 165-183 (2008).
- Wang, B. *et al.* How to Measure the Strength of the East Asian Summer Monsoon. *J. Climate* 21, 4449-4463 (2008).
- Wang, P.X. *et al.* The global monsoon across time scales: Mechanisms and outstanding issues. *Earth-Sci. Rev.* **174**, 84-121 (2017).
- Wang, B. Climatic regimes of tropical convection and rainfall. *J. Climate* 7, 1109-1118 (1994).
- Webster, P. J. The coupled monsoon system. In Bin Wang (editor), *The Asian Monsoon*, Springer, p. 3-66 (2006).
- Yan, Q., Zhang, Z. & Gao, Y. An East Asian Monsoon in the Mid-Pliocene. *Atmospheric and Oceanic Science Letters* **5**, 449-454 (2012).
- Zhang, R., Yan, Q., Zhang, Z., Jiang, D., Otto-Bliesner, B. L., Haywood, A. M., Hill, D. J., Dolan, A. M., Stepanek, C., Lohmann, G., Contoux, C., Bragg, F., Chan, W.-L., Chandler, M. A., Jost, A., Kamae, Y., Abe-Ouchi, A., Ramstein, G., Rosenbloom, N. A., Sohl, L., Ueda, H., 2013. Mid-Pliocene East Asian monsoon climate simulated in the PlioMIP. *Clim. Past* 9, 2085–2099.
- Zhang, Z., Sun, X. & Yang, X. Understanding the Interdecadal Variability of East Asian Summer Monsoon Precipitation: Joint Influence of Three Oceanic Signals. *J. Climate* **31**, 5485-5506 (2018).
- Zhao, G. *et al.* A New Upper-Level Circulation Index for the East Asian Summer Monsoon Variability. *J. Climate* **28**, 9977-9996 (2015).

----- (The end)

REVIEWERS' COMMENTS:

Reviewer #1 (Remarks to the Author):

As stated in my previous reviews, this paper provides the first phytolith-based record of vegetation change in eastern Eurasia (specifically China) during the late Miocene-Pleistocene. The data presented by the Wang et al. convincingly demonstrates a shift from C4 (or PACMAD) dominated grasslands to pooid-dominated grasslands at ~4 Ma, pointing to cooling and drying during this time. These results should be of interest to a wide audience of biologists and geologists.

The paper is well written and researched and very appropriate for Nature Communications. After reviewing the manuscript again, I found that the authors had addressed all my previous comments.

I only have one further comment, namely to recommend that the authors change the following sentence in the abstract: "Our newly obtained records conflict with the landform growth forcing hypothesis;..." The "landform growth forcing hypothesis" is not commonly known, and since Nat Comm is a broad journal, it would be better to use something that indicates with landform they are talking about. For example, "Our newly obtained records conflict with the hypothesis that the growth of the Tibetan Plateau influenced..."

Caroline Strömberg

Reviewer #2 (Remarks to the Author):

Dear authors,

thank you very much for addressing my points. As regards this ms, I don't have any further suggestions.

(line 161 in the new text: "..has fallen")

Best,
Torsten Utescher

Reviewer #4 (Remarks to the Author):

This seems like a nice careful and potentially high impact study of the evolving climate in northern China since the late Miocene using a new proxy method which should be of quite wide interest within the paleoclimate community studying the development of the Asian monsoon. There is generally a lack of good proxy data constraining how the rainfall has evolved over long periods of time especially for the continental areas and the community is presently too dependent on oceanic records so I feel that this study should be of wide interest. I didn't follow the logic in a few places which I highlight in the attached specific comments but I don't feel that there is any intrinsic major problem with this paper. I think that a good effort to address the points that I have raised below would result in a high quality study. In many ways the paper is reinforcing what we already suspect about the development of the East Asian monsoon and perhaps a few comments about the fact that this trend agrees with some inferred from marine sediments and that the climatic control inferred is not an entirely novel finding would perhaps make this paper look a bit more honest but I still feel that it is a useful contribution and I recommend publication after moderate revision

Line 23 decrease in Asian monsoon rainfall in the Pliocene - Could this reflect a change in seasonality instead?

Line 35 - increased warm-season precipitation - Not many people agree with that anymore. Not even Quade, More C4 is drier

Line 43 - suggested a reduction in warm-season precipitation in East Asia at ~4.0 Ma - Yes also see marine records

Wan, S., Li, A., Clift, P.D., Stuut, J.-B.W., 2007. Development of the East Asian monsoon: Mineralogical and sedimentologic records in the northern South China Sea since 20 Ma. *Palaeogeography, Palaeoclimatology, Palaeoecology*, 254(3-4): 561–582.

Clift, P.D., Wan, S., Blusztajn, J., 2014. Reconstructing Chemical Weathering, Physical Erosion and Monsoon Intensity since 25 Ma in the northern South China Sea: A review of competing proxies. *Earth-Science Reviews*, 130: 86-102. doi:10.1016/j.earscirev.2014.01.002.

Line 82 uneven distribution – uneven in time?

Line 87 - soil B horizons - What does “B” mean?

Line 99 - markedly decreases by ~4.2 Ma - markedly decreased by ~4.2 Myr ago

Line 100 - distinct decrease in monsoon precipitation - I think something needs to be said about seasonality

Line 116 dominated the grassland in China - You mean in the past? Or now?

Line 127 - However, this quantitative reconstruction is valid only - Do you make that assumption or not?

Line 147 - when Pooideae grasses continued to dominate - You mean the Holocene or also the LGM?

Line 156 - coexistence approach (CA) - You use too many abbreviations. They are confusing.

Line 159 - high temperatures and precipitation - You mean MAP? Or seasonal monsoonal?

Line 160 – WMMP - I already forgot what this means

Line 161 – WSP - Too many

Line 166 - than that in the next stage - Perhaps you should just say what that is

Line 179 - early Miocene sediments - Lower Miocene sediments

Line 180 Shanwang in northern China - Show this on a map if you are going to be specific

Line 198 - a similar vegetation pattern was discovered in North America - Is this relevant?

Line 205 – Lingtai – Label on map

Line 224 - -8~9‰ - -8 to -9‰

Line 236 - may date back to the early Miocene - Presumably to the establishment of the modern monsoonal climate zonation.

Line 258 - a shift to a dry and cold climate in the late Pliocene - Across all east Asia or just the study area?

Line 277 - 4 sites -Four sites - Spell out whole numbers up to ten

Line 306 - inconsistent with the late Miocene cooling - Perhaps the local climate is different?

Responses to comments to

Wang et al. “Asian monsoon rainfall variation during the Pliocene forced by global temperature change”

October 22, 2019. Nanjing.

Reviewer #1

I only have one further comment, namely to recommend that the authors change the following sentence in the abstract: "Our newly obtained records conflict with the landform growth forcing hypothesis;..." The "landform growth forcing hypothesis" is not commonly known, and since Nat Comm is a broad journal, it would be better to use something that indicates with landform they are talking about. For example, "Our newly obtained records conflict with the hypothesis that the growth of the Tibetan Plateau influenced..."

Response: Thank you. We have revised this sentence to: “Our newly obtained records conflict with the hypothesis that the growth of the Tibetan Plateau strengthened the Asian monsoon rainfall. Nevertheless, they emphasize the importance of global temperature as a determinant of Pliocene Asian monsoon variations.”

Reviewer #2

Line 161 in the new text: "...has fallen".

Response: Thank you. This change has been made.

Reviewer #4

1) Line 23 decrease in Asian monsoon rainfall in the Pliocene - Could this reflect a change in seasonality instead?

Response: Thank you. It is indeed possible. The decrease of the mean annual precipitation (MAP) could indicate a change in seasonality, because the MAP was mainly determined by summer rainfall, a decrease of MAP might result in a decrease in the difference between summer and winter precipitation, which is thus a seasonality change. Furthermore, we added data of the warmest month precipitation (WMMP) and the warm season (April to September) precipitation (WSP), which are indicators of the monsoon precipitation, to reconstruct the Asian monsoon variations (please see Supplementary Table 5). The synchronous decrease in the mean annual precipitation, the warmest month precipitation and the warm season precipitation suggest that a decrease in Asian monsoon rainfall instead of a change in seasonality. We have discussed this in the manuscript, line 251~258, page 12.

2) Line 35 - increased warm-season precipitation - Not many people agree with that anymore. Not even Quade, More C4 is drier

Response: Thank you. We agree to that the question of an increase warm-season precipitation need to be tested, and this is why we listed these controversial results. We found that a major reason for the discrepancy is the use of stable carbon isotopic compositions as a proxy for the expansion of C4 grasses and therefore monsoon precipitation, while the drivers of C4 expansion and the environmental niches of C4 grasses are still unclear in East Asia. As an alternative to stable carbon isotope ($\delta^{13}\text{C}$) analysis, our fossil phytoliths can provide detailed information on C3 (forests/shrubs/C3 grasslands) and C4

(Chloridoideae-dominated/Panicoideae-dominated grasslands) vegetation, thus providing detailed information on climate change.

In addition, it is true that more C4 means drier in the tropical and subtropical region, but this is not always the case in the temperate climate with limit precipitation, such as Chinese Loess Plateau where there are also abundant of C3 grasses (adapted to colder and drier environment than C4). The limited factors to the growth of C4 grasses in this semiarid region varied.

3) Line 43 - suggested a reduction in warm-season precipitation in East Asia at ~4.0 Ma - Yes also see marine records

Wan, S., Li, A., Clift, P.D., Stuut, J.-B.W., 2007. Development of the East Asian monsoon: Mineralogical and sedimentologic records in the northern South China Sea since 20 Ma. *Palaeogeography, Palaeoclimatology, Palaeoecology*, 254(3-4): 561–582.

Clift, P.D., Wan, S., Blusztajn, J., 2014. Reconstructing Chemical Weathering, Physical Erosion and Monsoon Intensity since 25 Ma in the northern South China Sea: A review of competing proxies. *Earth-Science Reviews*, 130: 86-102. doi:10.1016/j.earscirev.2014.01.002.

Response: Thank you very much for these. We cited Clift et al., 2014, *Earth-Science Reviews* in our manuscript, Ref. 42, line 271, page 13.

4) Line 82 uneven distribution – uneven in time?

Response: Yes, we mean uneven distribution in time. Corrected.

5) Line 87 - soil B horizons - What does “B” mean?

Response: the soil B horizon means a horizon that is formed below the surface mineral horizon (A) or organic horizon (O) in soil profile, a layer with accumulation of Fe, Al, organic matter, clay, salts, or carbonates. We use “accumulation” instead of “B” in the new version of the manuscript.

6) Line 99 - markedly decreases by ~4.2 Ma - markedly decreased by ~4.2 Myr ago

Response: This change has been made.

7) Line 100 - distinct decrease in monsoon precipitation - I think something needs to be said about seasonality

Response: Thank you. The final paragraph of the introduction section is a summary of major results and conclusions, we have discussed seasonality in the latter part of the manuscript (line 251~258, page 12). Please also see our response to your first comment.

8) Line 116 dominated the grassland in China - You mean in the past? Or now?

Response: Now. We add “modern” in the manuscript.

9) Line 127 - However, this quantitative reconstruction is valid only - Do you make

that assumption or not?

Response: We are sorry for the vague sentence. We rewrote it as “However, in order to use this regional scale grass habitat for quantitative reconstructing, we need to make the assumption that the same species have thrived since the late Miocene and that the environmental niches of these grass subfamilies have not evolved.”

10) Line 147 - when Pooideae grasses continued to dominate - You mean the Holocene or also the LGM?

Response: We mean the interval of 130~0 ka, including Holocene, the last glacial and interglacial periods. Please see Supplementary Table 2 for ages of our samples.

11) Line 156 - coexistence approach (CA) - You use too many abbreviations. They are confusing.

Response: We are sorry for the confusing. We replaced most of the abbreviations in the manuscript.

12) Line 159 - high temperatures and precipitation - You mean MAP? Or seasonal monsoonal?

Response: We mean both the MAP and the seasonal precipitation. Please see the corrected text: “Firstly, a low percentage of Pooideae grasses, together with the presence of Bambusoideae and Oryzoideae during 11.0~4.2 Ma, indicated a warm and humid climate during the late Miocene. The reconstructed MAT and MAP were 11~15.3°C and 800~1673 mm, respectively. The warmest month mean precipitation and the warm season (April to September) precipitation (WSP, which have fallen in

the monsoon season and thus are indicators of monsoon precipitation) were 76~199 mm and 427~1197 mm, respectively”. We also discussed seasonality in the manuscript, line 251~258, page 12.

13) Line 160 – WMMP - I already forgot what this means

Response: We are sorry for causing confusion. Revised.

14) Line 161 – WSP - Too many

Response: Corrected.

15) Line 166 - than that in the next stage - Perhaps you should just say what that is

Response: Revised.

16) Line 179 - early Miocene sediments - Lower Miocene sediments

Response: Corrected.

17) Line 180 Shanwang in northern China - Show this on a map if you are going to be specific

Response: This change has been made.

18) Line 198 - a similar vegetation pattern was discovered in North America - Is this relevant?

Response: Thank you for the comment. The vegetation structure prior to the expansion of C4 grasses is of great importance for the interpretation of climate change. If it was closed forest, more C4 means drier, such as South Asia; if it was open grassland, more C4 probably means more humid, such as East Asia and North America. This description also demonstrates phytoliths are very useful in reconstructing the vegetation structure, thus past environment change.

19) Line 205 – Lingtai – Label on map

Response: This change has been made.

20) Line 224 - -8~-9‰ - -8 to -9‰

Response: Corrected.

21) Line 236 - may date back to the early Miocene - Presumably to the establishment of the modern monsoonal climate zonation.

Response: Yes, this could indicate the establishment of the modern monsoonal climate. The reason we referenced Cave et al., 2016, Geology is that this study interpreted the change of $\delta^{13}\text{C}_{\text{SC}}$ as the change of precipitation and productivity, thus soil respiration flux, which could be the solution for the inconsistency between our

data and $\delta^{13}\text{C}_{\text{SC}}$ data. The recent paper presented by Cave et al., 2018, Earth-Science Reviews, using more $\delta^{18}\text{O}$ and $\delta^{13}\text{C}$ data, they draw the conclusion that atmospheric circulation has remained similar to today for at least past 55 million years.

22) Line 258 - a shift to a dry and cold climate in the late Pliocene - Across all east Asia or just the study area?

Response: Thank you for the comments. These fossil mammal data were compiled from a large area of East Asia, thus we could say this shift to a dry and cold climate happened in the late Pliocene across large area of East Asia. We add “across large area of East Asia” in the manuscript.

23) Line 277 - 4 sites -Four sites - Spell out whole numbers up to ten

Response: Corrected.

24) Line 306 - inconsistent with the late Miocene cooling - Perhaps the local climate is different?

Response: Yes, we agree with you that the regional data are not necessarily consistent with late Miocene cooling suggested by global SST. However, we did not obtain enough data to describe this relatively warm and humid period during the late Miocene to early Pliocene. More works, such as high resolution records, would be needed to reach a solid conclusion.

----- (the end)